# Online Detection for Black-Box Large Language Models with Adaptive Prompt Selection

## Abstract

The widespread success of large language models (LLMs) has made them integral to various applications, yet security and reliability concerns are growing. It now becomes critical to safeguard LLMs from unintended changes caused by tampering, malicious prompt injection, or unauthorized parameter updates, etc. Early detection of these changes is essential to maintain the performance, fairness, and trustworthiness of LLM-powered applications. However, in black-box settings, where access to model parameters and output probabilities is unavailable, few detection methods exist. In this paper, we propose a novel online change-point detection method for quickly detecting changes in black-box LLMs. Our method features several key innovations: 1) we derive a CUSUM-type detection statistic based on the entropy and the Gini coefficient of the response distribution, and 2) we utilize a UCB-based adaptive prompt selection strategy for identifying change-sensitive prompts to enhance detection. We evaluate the effectiveness of the proposed method using synthetic data, where changes are simulated through watermarking and model version updates. Our proposed method is able to detect changes quickly while well controlling the false alarm rate. Moreover, for real-world data, our method also accurately detects announced changes in LLM APIs via daily online interactions with APIs. We also demonstrate strong evidence of unreported changes in APIs, which may be of independent interest.

## 1 Introduction

Large Language Models (LLMs) have emerged as a transformative force in the field of artificial intelligence, demonstrating remarkable capabilities across a wide range of applications, from healthcare and finance to education and creative industries (Zabir & Peng, 2024; Lee et al., 2024; Moore et al., 2023; Çelen et al., 2024). LLMs are now integral components of chatbots, virtual assistants, and automated customer service systems (Dam et al., 2024; Dong et al., 2023; Pandya & Holia, 2023). Moreover, they're increasingly used in complex decision-making processes, e.g., LLM agents can interpret commands, make decisions, and take actions based on natural language inputs so as to assist in task planning, problem-solving, and automate certain workflows in software development or data analysis (Alshahwan et al., 2024; Hong et al., 2024; Eigner & Händler, 2024). This widespread integration is revolutionizing how businesses and individuals interact with information and technology, making LLMs a cornerstone of modern AI-driven solutions.

Despite their undeniable potential, the widespread adoption of LLMs has given rise to various safety, reliability, and consistency concerns (Bommasani et al., 2021; Biswas & Talukdar, 2023). As LLMs become increasingly embedded in critical systems, the risks associated with their vulnerabilities and stability become more pronounced. LLM-powered applications are susceptible to various threats, such as unauthorized model parameter updates and malicious prompt injections by hackers (Kang et al., 2024; Wu et al., 2024). These security issues can lead to shifts in the output distributions of LLMs, causing the generation of misleading and harmful content (Chao et al., 2023), or leakage of sensitive customer information (Ayyamperumal & Ge, 2024). Throughout the paper, we term shifts of LLMs' output distributions as *changes*. However, not all changes in LLM output distributions are necessarily harmful. Even benign changes, such as those introduced by LLM version updates and patches, can influence their output distributions, potentially rendering inconsistent behaviors before and after the change (Echterhoff et al., 2024). For example, Chen et al. (2024) thoroughly analyzed behavior drifts in GPT-3.5 and GPT-4 over time (March and June) across diverse tasks, including mathematical reason-

ing and opinion surveys. Moreover, the use of watermark without users' knowledge also infringes on users' right to be informed (Molenda et al., 2024). These concerns are particularly alarming. Timely detection of changes in LLMs allows for necessary intervention and ensures continued safety and reliability. See Figure 1 for an illustration of LLM changes and detection procedure.

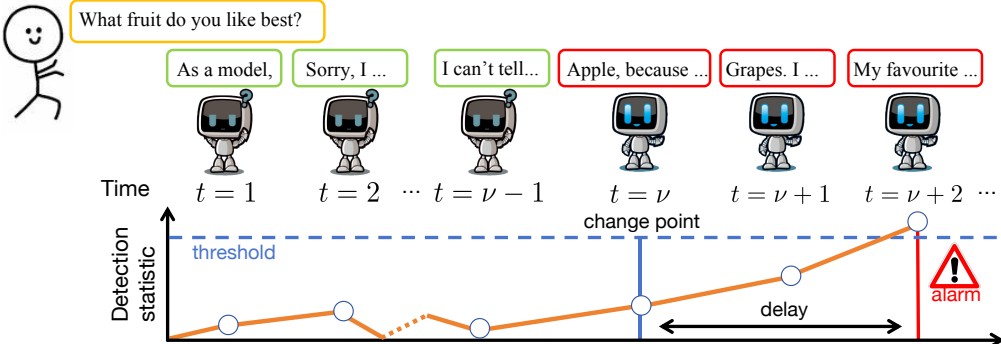

Figure 1: The detector interacts with an LLM in a sequence of time steps by collecting responses to a prompt. From some unknown change point $\nu$, there is a change in the LLM. The detection statistic of the detector keeps growing after $\nu$ until it hits the threshold to raise an alarm.

While the need for effective detection of changes in LLMs is clear, existing approaches face significant limitations, particularly in real-world scenarios due to two major challenges. Firstly, most existing detection techniques are designed for white-box models (Tang et al., 2024), assuming full access to model parameters and output probabilities. However, many LLMs operate as black-box systems, with the internal workflows opaque to users and operators. Secondly, we consider change detection in the online setting, where one needs to dynamically interact with an LLM using prompts and collect generated responses. Nonetheless, most existing methods focus on the offline setting, where pre-collected data is available and the goal is to devise a hypothesis testing framework or train a classifier for identification, such as determining the existence of watermark in a generated text (Gloaguen et al., 2024; Wu et al., 2023). In this regard, we pose the following question:

*How can we swiftly and accurately detect changes in black-box LLMs in online settings?*

In this paper, we propose a novel online change detection method specifically designed for black-box LLMs. We consider deploying a detector dynamically interacting with an LLM in a sequential manner. In each round, the detector queries (selected) prompts and collects responses from the LLM. To tackle the aforementioned challenges, our approach features the following innovations. Firstly, we derive a CUSUM-type detection statistic that is updated sequentially based on newly collected responses. This is a variant of the seminal CUSUM test (Page, 1954) to handle unknown distributions, and is derived in a way such that the statistic remains around zero before the change and increases linearly afterward. Thus the value of detection statistic indicates the likelihood of the emergence of a change. Secondly, we utilize entropy and Gini coefficient-motivated (Tang et al., 2023) quantities to characterize the distribution of responses, which avoids direct model inference on LLM. Besides, to boost the detection performance, we adopt a UCB-based adaptive prompt selection strategy to identify change-sensitive prompts, thereby optimizing the detection process.

We evaluate our detection algorithm in both synthetic and real-world environments. In synthetic scenarios, we simulate responses of LLMs transitioning from unwatermarked to watermarked and between different LLM versions. In real-world cases, we collect a streaming dataset composed of responses to 20 prompts using 9 LLM online APIs, spanning from June 1st to August 31st, 2024. We validate our algorithm on this dataset, successfully identify an officially confirmed change in the Mistral API (Mistral AI, 2024), and two unconfirmed changes in GPT-4 Turbo (OpenAI, 2024) and Jamba (AI21Lab, 2024) with strong evidence. We summarize our contributions as follows.

• We propose a recursively updated CUSUM-type detection statistic to effectively identify changes in LLMs. By utilizing entropy and Gini coefficient-inspired quantities, our method captures the variability in response distributions, making it well-suited for black-box LLMs.

• We propose a UCB-based strategy for dynamically selecting change-sensitive prompts during sequential interactions with LLMs. This approach improves detection efficiency by focusing on prompts that are more likely to reflect changes.

• We demonstrate the effectiveness of our approach through extensive numerical experiments, including synthetic environments and LLM online APIs. Our synthetic environments introduce watermarking and language model version changes as change points. Our detection approach accurately identifies these change points with well-controlled false alarm rates. When applied to LLM APIs, our approach locates an officially announced model update through limited daily queries on one LLM API. We also suggest probable unconfirmed changes with strong evidence.

**Related Work**    There are two lines of work closely related to our study.

*Detection in LLMs:*  Recent studies have primarily focused on detecting LLM-generated text and watermarked data in the offline setting; see Liu et al. (2024); Yang et al. (2023) for a comprehensive survey. Kirchenbauer et al. (2023) introduced a soft watermarking method that utilizes green and red lists alongside a detection algorithm based on hypothesis testing. Subsequently, numerous variants have been proposed to empirically enhance the trade-off between watermark detectability and text quality (Lu et al., 2024; Giboulot & Teddy, 2024; Hoang et al., 2024). At the theoretical level, Li et al. (2024) introduced a statistical framework for designing watermark and the guarantee on detection accuracies. Yet these detection methods operate in a white-box setting, requiring prior knowledge of the watermark scheme. In black-box settings, Gloaguen et al. (2024) proposed rigorous statistical tests to detect the presence of a watermark. Nevertheless, like most works, their approach primarily focuses on determining whether a given text originates from a watermarked LLM using a two-sample test, which is different from our online setting. Moreover, these methods are specific to certain types of watermarks and are not easily adaptable to other types of changes.

*Online Change Detection Methods:*  The problem of online change detection has been extensively studied in statistics and signal processing, see Poor & Hadjiliadis (2008); Tartakovsky et al. (2014) for summaries of earlier work. Our proposed method is primarily inspired by the cumulative sum (CUSUM) test Page (1954). The core idea of the CUSUM test is to accumulate the log-likelihood ratio, which has a negative mean in the pre-change regime and a positive mean in the post-change regime. For unknown and non-parametric distributions, one approach has been to estimate the log-likelihood ratio and the CUSUM statistic using pre-collected training datasets. This includes methods such as kernel estimation (Kawahara & Sugiyama, 2009), neural network estimation (Moustakides & Basioti, 2019), and density estimation (Liang & Veeravalli, 2024). Another approach is to replace the log-likelihood ratio with some other useful statistic for distinguishing between distributions in constructing tests. Examples of these approaches include the use of kernel M-statistics (Li et al., 2015), one-class SVMs (Desobry et al., 2005), nearest neighbors (Chen, 2019), and Geometric Entropy Minimization (Kurt et al., 2020). However, none of these methods are suitable for black-box LLMs due to the large cardinality of the token set and the need for computational efficiency in online settings. To address this, we replace the log-likelihood ratio with the deviation-to-nominal quantities of our entropy and Gini statistics in developing our detection procedure.

## 2   PROBLEM SETUP: ONLINE CHANGE DETECTION FOR LLMS

Recall that we refer to changes as shifts in the output distributions of LLMs. To detect these changes, we deploy a detector sequentially interacting with LLMs by querying input prompts and collecting generated responses. We denote input prompt as $x \in \mathcal{X}$ and the generated responses as $Y = \{y^1, \ldots, y^C\}$. Here $\mathcal{X}$ is the set of possible prompts, $C$ is a constant, and $y^1, \ldots, y^C$ are independently generated responses to the same input prompt. Equivalently, we view $y^1, \ldots, y^C$ as i.i.d. samples from the conditional distribution $P(\cdot|x)$ parameterized by an LLM. The repeated responses provide sufficient information of the output distributions of the LLM. To ease the presentation, we drop the superscript of repetition index on response $y$ when there is no confusion. Each response consists of a sequence of words called tokens. We denote $z$ as a token, and for a response $y$ with $\ell$ tokens, we have $y = \{z_1, \ldots, z_\ell\}$. Each token is chosen from a finite token set $\mathcal{V}$.

At the $t$-th round of interaction between the detector and an LLM with $t \in \mathbb{N}_+$, $K$ distinct query prompts $\{x_{t,1}, \ldots, x_{t,K}\}$ are sent to the LLM and the corresponding responses $\{Y_{t,1}, \ldots, Y_{t,K}\}$ are collected. We assume the responses are uncorrelated with each and past data. This is to ensure that the LLM is not adapting to our queries. We can achieve this by only querying the LLM with the current prompt $x$ without historical conversation. In the presence of a change point, the online responses are generated following the scheme, for any $k \in [K]$ and $y_{t,k}^c \in Y_{t,k}$,

$$y_{t,k}^c \sim P_0(\cdot|x_{t,k}), \quad \text{for } t = 1, 2, \ldots, \nu - 1,$$
$$y_{t,k}^c \sim P_1(\cdot|x_{t,k}), \quad \text{for } t = \nu, \nu + 1, \ldots,$$

where $\nu$ is an unknown change point, and both $P_0$ and $P_1$ are unknown. It is worth mentioning that the difference between $P_0(\cdot|x)$ and $P_1(\cdot|x)$ varies depending on the input prompt $x \in \mathcal{X}$: Some prompts lead to appealing distinguishability, yet some may even yield $P_0(\cdot|x)$ and $P_1(\cdot|x)$ identical.

Our task is to identify the unknown change point $\nu$ as quickly as possible while controlling the false alarm rate, i.e., the probability of incorrectly raising alarm when there is no change. Hypothetically, the change point $\nu$ can occur at any time, but an early change is of less interest especially when we do not have prior knowledge of $P_0$ and $P_1$. In that case, for the majority of time steps, we operate under $P_1$ without a change. Therefore, we focus on the scenario in which $\nu$ is relatively large and we always assume we have adequate time for accumulating information of $P_0$ via interactions before our detection procedure starts. This assumption, which presumes the availability of data from the pre-change regime, is common in online detection problems and is often the case in applications (Yu et al., 2023). We view the data collected prior to the detection procedure as *historical data*. Such historical data consists of prompts in $\mathcal{X}$ and their corresponding responses, which help the detector distinguish new data collected after the detection procedure starts.

**Query Budget** During interaction with an LLM, we have a *query budget $K$*, arising from two reasons. First, the cardinality of the prompt set $\mathcal{X}$ is usually large, making it computationally infeasible to exhaustively query every prompt at each round. Second, different prompts exhibit varying sensitivity to a certain change. Prompts of high sensitivity tend to detect the changes quickly, but they are unknown in advance. Therefore, we aim to enhance the detection performance by actively selecting prompts at each round of interaction, based on all historical data. In other words, our goal is to select the most sensitive prompts to accelerate the detection process.

**Performance Criteria** The detector identifies a change point by returning a stopping time $T$ based on collected data. We use two common criteria to measure the performance, Average Detection Delay (ADD) and Average Run Length (ARL). ADD is the average delay between the stopping time $T$ and the true change-point $\nu$, and a smaller ADD indicates faster detection. ARL measures the expectation of $T$ when no change occurs, thus a larger ARL implies a lower false alarm rate.

## 3 DETECTION ALGORITHM

We present the proposed online algorithm for change detection in black-box LLMs, as depicted in Figure 2. Our algorithm consists of two building modules: 1) a detection module with a given query prompt in subsection 3.1 and 2) a selection module for screening change-sensitive prompts in subsection 3.3. We introduce them in order and then combine them to derive our detection algorithm.

### 3.1 DETECTION WITH A GIVEN PROMPT

Recall that we denote $y$, tokenized as $y = \{z_1, \ldots, z_\ell\}$, as a randomly generated response to a given prompt $x$, i.e., $y \sim P(\cdot|x)$. We aim to determine if $P(\cdot|x)$ changed at some time. Although it is tempting to estimate $P(\cdot|x)$ directly, it is intractable due to the enormous size of vocabulary. Classical detection methods such as likelihood ratio statistics are not applicable either. Instead, we resort to the entropy and Gini coefficient-based metric to distinguish distributions. To further reduce the computational overhead, we only consider the joint distribution of first $N$ ($N \leq \ell$) tokens. In the extreme case, we allow $N = 1$.

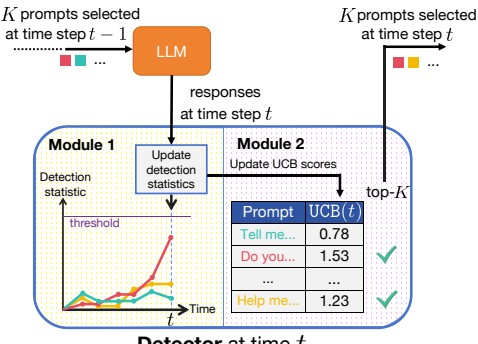

Figure 2: Flowchart of our detection algorithm. At round $t$, the detector uses $K$ prompts selected at time $t-1$ to query the LLM and updates the detection statistics (Module 1) of the selected prompts. The detector then updates the UCB scores (Module 2) and select prompts with the top-$K$ UCB scores to be queried at next round.

**Entropy-Based Metric** We begin with the extreme case of $N = 1$ and then extend to $N > 1$. The distribution of $z_1$ given $x$ is known as the Next Token Probability (NTP), whose entropy, termed as *first-token entropy*, is defined as

$$\texttt{FTE}(x) = -\sum_{z_1 \in \mathcal{V}} \mathbb{P}(z_1|x) \log \mathbb{P}(z_1|x).$$

In implementation, we approximate $\mathbb{P}(z_1|x)$ by its empirical version. Details for implementation are provided in subsection 3.2. Entropy is particularly suitable for limited empirical data, as it

features low variance and low stochastic error (Paninski, 2003). Unfortunately, not all changes can be effectively captured by the distribution of the first token. Therefore, we enlarge the token length to $N$ and define the following *N-token entropy*, which is denoted as NTE:

$$\text{NTE}(x) = -\sum_{\{z_1,\dots,z_N\}\in\mathcal{V}^N} \mathbb{P}(z_1,\dots,z_N|x)\log\mathbb{P}(z_1,\dots,z_N|x).$$

We discuss how to approximate NTE with empirical data in subsection 3.2.

**Gini Coefficient-Based Metric** Entropy exhibits high sensitivity to how probability mass is spread out among all possible outcomes (Arnez et al., 2024). On the other hand, Gini coefficient is sensitive to changes of dominant outcomes in a distribution, and it has good performance in watermark detection (Tang et al., 2023). Thus, we adopt Gini coefficient to complement entropy for better detection. Similar to entropy-based metric, we begin with the case for the first token, termed *first-token Gini*. The *first-token Gini* metric $\text{FTG}(x)$ is defined as

$$\text{FTG}(x) = 1 - \frac{1}{|\mathcal{V}|}\sum_{i=1}^{|\mathcal{V}|}(F_i + F_{i-1}),$$

where we let $p_{(i)}, i = 1, 2, \dots, |\mathcal{V}|$ be the probabilities $\{\mathbb{P}(z_1|x), z_1 \in \mathcal{V}\}$ sorted in ascending order, and $F_i = \sum_{j=1}^{i} p_{(j)}$ is the cumulative probability up to the $i$-th smallest value. We define $F_0 = 0$. Derivations of FTG for discrete distributions are detailed in Appendix A. Similarly, the *N-token Gini* for the first $N$ tokens is computed as

$$\text{NTG}(x) = 1 - \frac{1}{|\mathcal{V}|^N}\sum_{i=1}^{|\mathcal{V}|^N}(F_i + F_{i-1}),$$

where we reload $p_{(i)}$ as the $i$-th smallest probability of $\mathbb{P}(z_1,\dots,z_N|x)$ while $\{z_1,\dots,z_N\}$ taking values in $\mathcal{V}^N$, and then $F_i$ is defined the same as that in *first-token Gini*. We also defer efficient computation of NTG to subsection 3.2.

**Detection Statistic and Procedure** For each of the four metrics, we propose to aggregate their *deviations from historical value* in each interaction round to derive a cumulative sum type detection statistic. We take the *first-token entropy* as an example. In round $t$, we calculate $\text{FTE}(x)$ using newly collected data within this round. Then the detection statistic $W_{\text{FTE}}(t; x)$ is updated by

$$W_{\text{FTE}}^{+}(t;x) = \max\{0, W_{\text{FTE}}^{+}(t-1;x) + (\text{FTE}(x) - \mu_{\text{FTE}}(x)) - d_{\text{FTE}}\},$$
$$W_{\text{FTE}}^{-}(t;x) = \max\{0, W_{\text{FTE}}^{-}(t-1;x) - (\text{FTE}(x) - \mu_{\text{FTE}}(x)) - d_{\text{FTE}}\},$$
$$W_{\text{FTE}}(t;x) = \max\{W_{\text{FTE}}^{+}(t;x), W_{\text{FTE}}^{-}(t;x)\}.$$

Here, $W_{\text{FTE}}^{+}$ and $W_{\text{FTE}}^{-}$ monitor positive and negative shifts of the FTE values, $\mu_{\text{FTE}}$ is the average of $\text{FTE}(x)$ in historical data, and $d_{\text{FTE}}$ is a drift. The drift term is set properly to ignore minor stochastic deviations of $\text{FTE}(x)$ to its historical average. The detection statistic $W_{\text{FTE}}$ is expected to oscillate around zero during the pre-change rounds, but exhibits a positive drift in the post-change rounds if the FTE values differ before and after the change. This behavior mimics the seminal CUSUM statistic (Page, 1954). Due to such properties, our detection statistics are capable of distinguishing the post-change data from the pre-change data. The detection statistics using other metrics are defined in the same way, as summarized in Module 1, where we unify the notation by denoting $s$ as a string in metrics set $S = \{\text{FTE}, \text{FTG}, \text{NTE}, \text{NTG}\}$ and $W_s$ being one of $\{W_{\text{FTE}}, W_{\text{FTG}}, W_{\text{NTE}}, W_{\text{NTG}}\}$. We term $\{W_{\text{FTE}}, W_{\text{FTG}}, W_{\text{NTE}}, W_{\text{NTG}}\}$ as our *detection statistics*.

---

**Module 1** Detection_Statistics$(x, t)$: Update detection statistics for prompt $x$ at time $t$.

---

1: **Require**: Prompt $x$, index $t$.
2: **Parameter**: Historical mean value $\mu_s(x)$, drift parameter $d_s$, repetition time $C$, token length $N$, initial values $W_s(0; x) = 0$ for all $x \in \mathcal{X}$, detection statistics $W_s(t - 1; x)$ at time $t - 1$.
3: $Y_t \leftarrow$ sample black-box model using prompt $x$ for $C$ times, tokenize and truncate to $N$.
4: **for** metric $s$ in $S = \{\text{FTE}, \text{FTG}, \text{NTE}, \text{NTG}\}$ **do**
5:    $s(x) \leftarrow$ calculate corresponding metric from $Y_t$,
6:    $W_s^{+}(t;x) \leftarrow \max(0, W_s^{+}(t-1;x) + (s(x) - \mu_s(x)) - d_s)$,
7:    $W_s^{-}(t;x) \leftarrow \max(0, W_s^{-}(t-1;x) - (s(x) - \mu_s(x)) - d_s)$,
8:    $W_s(t;x) \leftarrow \max(W_s^{+}(t;x), W_s^{-}(t;x))$.
9: **end for**

---

Determining a change point in LLMs is now achieved by comparing the detection statistics with a threshold $b$ and stopping at the first moment that we have sufficient evidence, i.e.,

$$T = \inf \left\{ t : \max\{W_{\texttt{FTE}}(t; x), W_{\texttt{FTG}}(t; x), W_{\texttt{NTE}}(t; x), W_{\texttt{NTG}}(t; x)\} \geq b \right\}.$$

This can be interpreted as a *parallel monitoring* scheme in which four detection statistics are tracked simultaneously. Such parallel monitoring is advantageous and more effective compared to relying on a single statistic, as the nature of the change is unknown and may leave some of these statistics unaffected after the change. The threshold $b$ is chosen to satisfy the false alarm requirement while maintaining sensitivity to change detection; it can usually be determined via simulation using pre-change data. We remark that the scale of different detection statistics can be quite distinct, thus some normalization is needed for choosing the threshold $b$. We discuss this in Section 3.2.

**Remark 1** *Note that entropy and Gini coefficient may remain unchanged when the underlying distribution shifts. We adopt these two metrics as they can be computed under black-box models, computationally feasible under the large vocabulary set, and are empirically sensitive to changes in most cases. Moreover, as more prompts are queried, the chance of entropy and Gini remaining unchanged across all prompts diminishes significantly. Our algorithm is designed to be plug-and-play, allowing for the integration of other statistics, such as perplexity, to further enhance detection.*

### 3.2 Implementation Details of Detection Statistics

For *first-token entropy* and *first-token Gini*, we can directly approximate the first token probability given a prompt $x$ by empirical data. However, the computation of NTE and NTG becomes less clear due to the exponential growth of different combinations of $N$ tokens. To overcome the computational overhead, we propose the following approximation method akin to data augmentation.

A response $Y_{t,k}$ consists of $C$ independent responses $\{y_{t,k}^1, \ldots, y_{t,k}^C\}$. Recall that for each response $y_{t,k}^c$, we use an LLM tokenizer to tokenize it to a sequence of tokens, as $\{z_{t,k,1}^c, \ldots, z_{t,k,\ell}^c\}$. For simplicity, we omit the subscripts $t$ and $k$ in $z$, as the responses are taken at the same time $t$ and for the same prompt index $k$. We denote $\{z_i^1, \ldots, z_i^C\}$ as the set of $i$-th token extracted from each response in $\{y_{t,k}^1, \ldots, y_{t,k}^C\}$. When calculating FTE and FTG, we replace the population probability $\mathbb{P}$ by the empirical counterpart obtained using the first tokens $\{z_1^1, \ldots, z_1^C\}$. However, for NTE and NTG, we adopt a different approach. We merge together the first $N$ tokens as $\{z_1^1, \ldots, z_1^C, \ldots, z_N^1, \ldots, z_N^C\}$ and calculate its empirical distribution, which is denoted as $\hat{P}_{1:N}(\cdot|x)$. Note that $\hat{P}_{1:N}(\cdot|x)$ is different from the joint distribution of the first $N$ tokens and is easy to compute. We substitute $\hat{P}_{1:N}$ into NTE and NTG to obtain their empirical approximations. Through our experiments, we find that setting $N = C = 20$ leads to appealing performances; see Section 4.

To fully implement Module 1, we also need to find the historical average $\mu$ and drift $d$ for a prompt $x$. We aim to set a unified drift $d$ and detection threshold $b$ for simplicity. However, the four metrics have different scale, and thus need normalization. Historical average $\mu$ is estimated using historical data, which is collected in the first few rounds, say 20 rounds, of interaction as we focus on relatively late changes. In real applications, we can gather historical data within a very short time period by frequently query LLMs. Note that the detection procedure only starts after historical data collection. On the historical data, we compute the four metrics in each round, and average over different rounds to obtain $\mu$. We also find the standard deviation $\sigma$ for the four metrics. More specifically, during detection, we normalize $\texttt{FTE}(x)$ by $\texttt{FTE}'(x) = \frac{\texttt{FTE}(x) - \mu_{\texttt{FTE}}(x)}{\sigma_{\texttt{FTE}}(x)}$ so that $\texttt{FTE}'(x)$ is approximately zero mean and has unit variance. Other metrics are also normalized. From now on, we denote $s(x)$ as the normalized metrics, and $W_s$ as the detection statistics computed based on the normalized metrics, for $s \in S$. As all metrics are now in the same scale and so are the detection statistics, we adopt a unified choice of the drift $d$ and threshold $b$ for all the detection statistics. After normalization, it is plausible to use the maximum of the four detection statistics to do detection, as

$$W(t; x) = \max\{W_{\texttt{FTE}}(t; x), W_{\texttt{FTG}}(t; x), W_{\texttt{NTE}}(t; x), W_{\texttt{NTG}}(t; x)\}. \tag{1}$$

When running the detection procedure with only one prompt, we compare $W(t; x)$ with the threshold $b$ after each update, and stop at the first time when $W(t; x)$ exceeds $b$.

### 3.3 Detection with Adaptive Selection of Prompts

Querying a single prompt can limit the detection power, thus we allow $K$ different prompt queries at each interaction round. Different prompts have varying sensitivity to a change, and we need

to actively select the most change-sensitive prompts. The difficulty lies in that we have no prior knowledge of the sensitivity of prompts. This requires balancing exploration and exploitation, i.e., providing sufficient exposure to different prompts yet identifying good ones early.

We adopt the Upper Confidence Bound (UCB) algorithm for prompt selection, which is a benchmark for multi-armed bandits and enjoys theoretical optimality (Sutton, 2018). Specifically, at time step $t$, for each prompt $x$ in $\mathcal{X}$, we calculate $W(t; x)$ using Eq. (1). Prompts whose $W$ exhibits a higher growth rate after change are preferred. We use the increment on $W$ between consecutive times to gauge the sensitivity of prompts: larger increment after the change occurs is preferred. Accordingly, we denote $U(t; x) = W(t; x) - W(t - 1; x)$ as the reward function and select prompts based on the UCB score, which is the estimated reward plus the confidence interval, as

$$\text{UCB}(t; x) = \hat{U}(t; x) + \sqrt{\frac{\alpha \ln t}{2n(t; x)}} \quad \text{with} \quad \hat{U}(t; x) = \frac{1}{n(t; x)} \sum_{\tau=1}^{t} U(\tau; x),$$

where $n(t; x)$ is the number of times $x$ is selected in the past $t$ time steps, $\alpha$ is the confidence level parameter and $\hat{U}(t; x)$ is the estimated reward. At time $t + 1$, $K$ prompts with the highest UCB scores will be selected. Specially at time 1, we select all prompts for initialization. The detailed selection strategy is presented in Module 2.

Combining Module 1 and 2, we present our online change detection algorithm for black-box LLMs in Algorithm 3. At time step $t + 1$, we query the $K$ prompts selected at time $t$ and update the corresponding detection statistics. Note that for initialization, we query every $x$ in $\mathcal{X}$ at time 1. For prompts not selected at time $t$, their detection statistics remain unchanged as in the previous time step. After all detection statistics get updated, the detector will raise alarm if any of the detection statistics is above the preset threshold $b$.

---

**Module 2** `TopK_UCB`$(K)$: Select top $K$ prompts to be queried at time $t + 1$.

1: **Require**: Query budget $K$, previous time step $t$.
2: **Output**: Set of selected prompts $\mathcal{Z}$.
3: **for** prompt $x$ in $\mathcal{X}$ **do**
4:     $\hat{U}(t; x) \leftarrow W(t; x)/n(t; x)$,   $\text{UCB}(t; x) \leftarrow \hat{U}(t; x) + \sqrt{\frac{\alpha \ln t}{2n(t; x)}}$.
5: **end for**
6: Return top-$K$ prompts with the highest $\text{UCB}(t; x)$ values from $\mathcal{X}$ as $\mathcal{Z}$.

---

**Algorithm 3** LLM Online Change Detection With Adaptive Selection of Prompts

1: **Require**: Prompt set $\mathcal{X}$, query budget $K$, threshold $b$.
2: **Output**: Stopping time $T$.
3: **Init**: $t \leftarrow 0$, all detection statistics $\leftarrow 0$, $\mathcal{Z} \leftarrow \mathcal{X}$.
4: **while** not return **do**
5:     $t \leftarrow t + 1$,
6:     **for** prompt $x$ in $\mathcal{Z}$ **do**
7:        Update $W_s(t; x)$ using `Detection_Statistics`$(t; x), s \in S$,
8:        $W(t; x) \leftarrow \max_{s \in S} W_s(t; x)$.
9:     **end for**
10:    **for** prompt $x$ in $\mathcal{X} \setminus \mathcal{Z}$ **do**
11:       $W(t; x) \leftarrow W(t - 1; x)$.
12:    **end for**
13:    **if** $\max_{x \in \mathcal{X}} W(t; x) \geq b$ **then**
14:       Return $T \leftarrow t$.
15:    **end if**
16:    $\mathcal{Z} \leftarrow$ `TopK_UCB`$(K)$.
17: **end while**

---

## 4 EXPERIMENTS

We conduct experiments on two types of synthetic data with changes simulated through watermarking and version updates (Section 4.1), and on real-world responses collected from various LLM

APIs (Section 4.2). The prompts used across all experiments are listed in Table 1 in Appendix C, and will be referenced by their index throughout the text.

## 4.1 ONLINE DETECTION FOR SYNTHETIC DATA

### 4.1.1 DETECTION WITH ONE PROMPT

**Detect Emergence of Watermark** We generate responses of the LLM `facebook/opt-125m` to prompt 12 in Table 1. Before the change point, no watermark is applied, while after the change, the soft watermark (Kirchenbauer et al., 2023) is applied to the generated responses. More details about the soft watermark are provided in Appendix B. We generate a set of pre-change data consisting of 20 time steps as historical data, which is used to compute the historical mean and variance of the detection statistics. All metrics are then normalized using these historical values as outlined in subsection 3.2. We set the number of repeated responses $C = 20$, token size $N = 20$, and drift parameter $d = 0.5$ in Module 1 unless otherwise specified. This configuration of $C$ and $N$ is chosen to achieve a low average detection delay while maintaining computational efficiency, see Appendix C. Figure 3 shows the evolutions of the four metrics (*first-token entropy*, *N-token entropy*, *first-token Gini* and *N-token Gini*) and their cumulative values used as detection statistics. As shown, all detection statistics are able to detect the presence of a relatively strong watermark quickly. Additional results for other prompts and varying watermark strengths are provided in Appendix D.1.

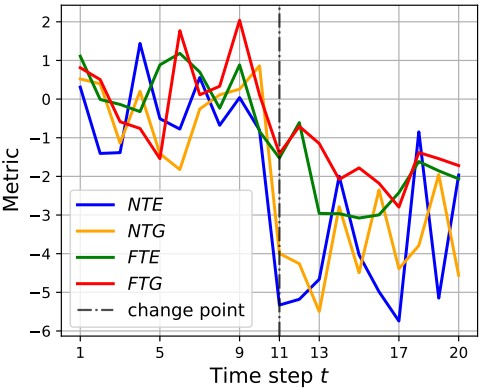 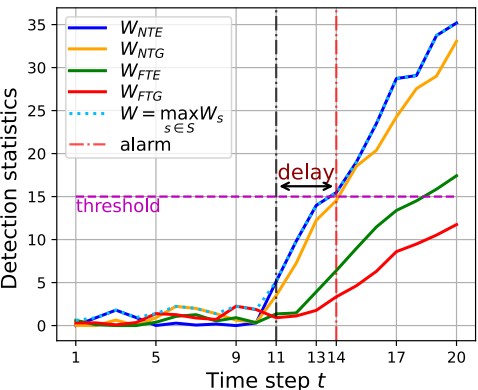

Figure 3: Evolution of the four metrics (Left) and their cumulative values used as detection statistics (Right), with post-change data generated via soft watermarking (with parameters $\delta = 2$, $\gamma = 0.5$) and change point $\nu = 11$. The four metrics show significant shifts after the change point. By applying the threshold shown in the right panel, the detection statistics raise an alarm at $T = 14$.

**Detect Synthetic Version Change** We synthesize three version change cases by setting one LLM as the pre-change model and one of its variants as the post-change. All models are available on Hugging Face. The query object is artificially switched from the pre-change model to the post-change model at a pre-set change point $\nu$. From the results shown in Figure 4, we observe that the detection statistics remain small before the change and exhibit linear growth after the change, enabling swift detection. For results on more prompts, see Figure 13 in Appendix D.1.

### 4.1.2 DETECTION WITH ADAPTIVE SELECTION OF PROMPTS

In this subsection, we focus on a specific *Version Change* from `facebook/opt-125m` to `facebook/opt-350m`, and perform the detection algorithm with adaptive prompt selection. The prompt set $\mathcal{X}$ consists of 14 prompts, indexed 0 to 13 in Table 1. We set the UCB parameter $\alpha = 8$ and select $K = 5$ prompts each time. To visualize the sensitivity of different prompts to the change, we plot the trajectories of detection statistics for individual prompts in Figure 5a. We then plot our detection statistics resulting from adaptive selection in Figure 5b, showing the algorithm effectively accumulates values from the most sensitive prompts, specifically prompt 8, 9, 10, 12, and 13 here.

To further illustrate the adaptive selection process, we plot the relative UCB scores in Figure 6a. Higher scores indicate a greater likelihood of selecting the corresponding prompt. After the change, the UCB scores of the most sensitive prompts dominate, enabling effective selection of these prompts. Additionally, we compare the ADD of our adaptive selection method with that under

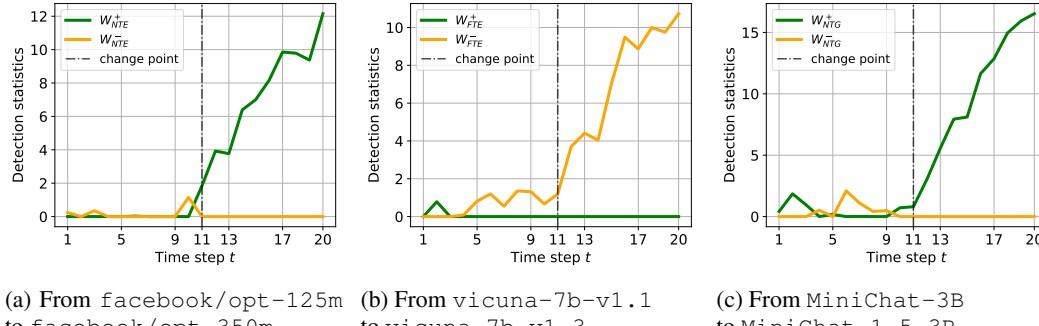

(a) From `facebook/opt-125m` to `facebook/opt-350m`  (b) From `vicuna-7b-v1.1` to `vicuna-7b-v1.3`  (c) From `MiniChat-3B` to `MiniChat-1.5-3B`

Figure 4: Detection statistics under three scenarios of version change, with change point set as $\nu = 11$. Both the positive branch ($W^+$) and negative branch ($W^-$) of selected detection statistics are shown. The prompts used in the three cases are prompts 10, 10, and 12, respectively. Since the various detection statistics exhibit similar trends, we use the best one for illustration.

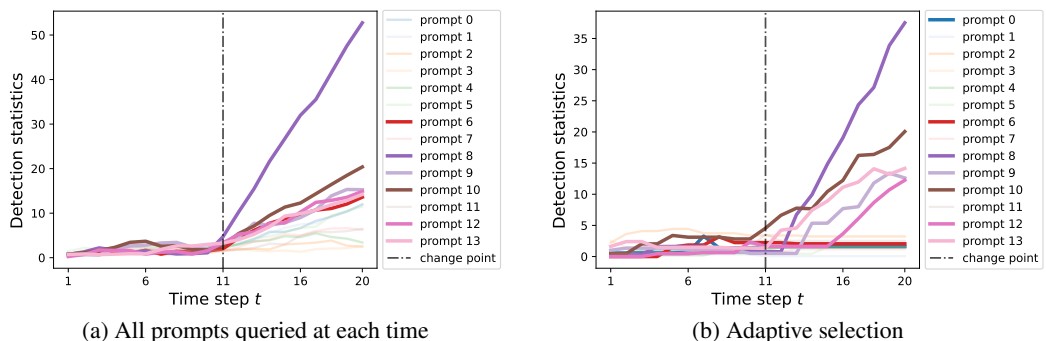

(a) All prompts queried at each time  (b) Adaptive selection

Figure 5: Trajectories of detection statistics (a): when every prompt is queried at each time step. In this case, prompts 8, 10, 9, 12, 13, 6 are the top six prompts with the highest growth rate after change, which are highlighted. (b): when we use our adaptive selection to select 5 prompts at each time step. It is shown that prompts with top growth rate are 8, 9, 10, 12, 13, which coincide with (a).

random selection as a baseline, and the ADDs using individual prompts, under various ARL levels, as shown in Figure 6b. The random strategy selects $K = 5$ prompts randomly from $\mathcal{X}$ at each time. Details on the simulation of detection thresholds for different ARLs are provided in Appendix C. After obtaining the threshold under a certain ARL, we repeatedly run the detection procedure and calculate the ADD. The results show that the ADD under our adaptive selection is smaller than that under random selection, and closely matches the best-performing individual prompt.

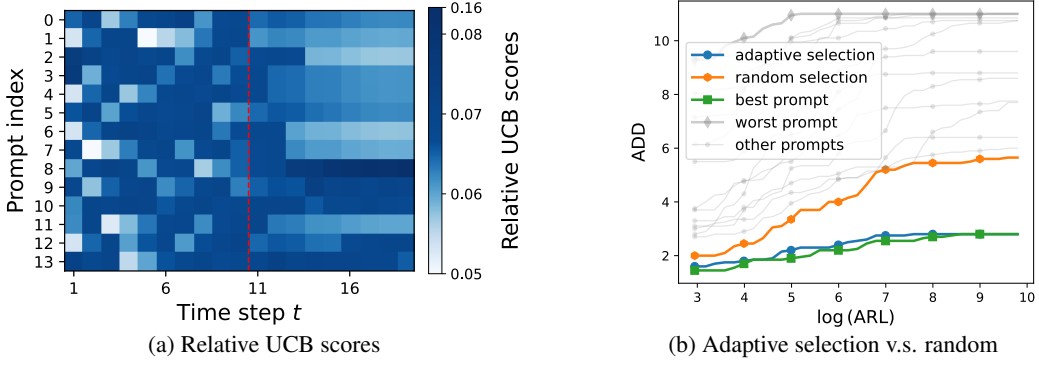

(a) Relative UCB scores  (b) Adaptive selection v.s. random

Figure 6: (a): Relative UCB scores – UCB score divided by the sum of UCB scores of all prompts. The change point $\nu = 11$ is marked in red. We can see the convergence on the most sensitive prompts (8, 9, 10, 12, and 13) after change. (b): ADD-ARL trade-off comparison between adaptive selection, random selection, and individual prompts. ADD is the average delay of 20 repeated experiments.

## 4.2 ONLINE DETECTION FOR REAL-WORLD APIS

We apply our proposed algorithm to real datasets collected by interacting with 9 LLM APIs: `gpt-4o`, `gpt-4`, `gpt-4-turbo`, `gpt-3.5-turbo` from OpenAI (2024), `command-r-plus` from Cohere (2024), `claude-3-haiku-20240307` from Claude (2024), `mistral-large-latest` from Mistral AI (2024) and `jamba-instruct`, `j2-ultra` from AI21 Labs (2024). We collected their responses once a day from June 1st, 2024, to August 31st, 2024, using 20 different prompts specified in Appendix C. Historical data were collected from June 1st to June 5th, 2024. We set the number of repeated responses $C = 100$ and token size $N = 20$. We use the tokenizer from `opt-125m` to tokenize the responses except for `command-r-plus` and `j2-ultra` which provide tokenization service.

Our detection procedure successfully detects a change that corresponds to an update of `mistral-large-latest` on July 24th, 2024, as confirmed by their website (Mistral AI, 2024). In Figure 7, we illustrate the detection statistic for *N-token Gini* using prompt 0. Similar patterns for other prompts are provided in Appendix D.3.

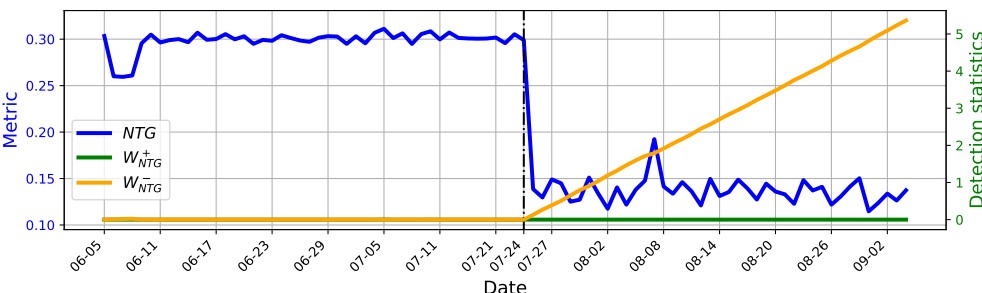

Figure 7: LLM API change detected in `mistral-large-latest` on July 24th, 2024, corresponding to an update officially announced by Mistral AI.

Furthermore, in certain instances, our detection statistics raise strong alarms, even in the absence of officially announced updates. These unconfirmed changes are mostly detected by only a subset of prompts. A possible explanation for this phenomenon is that the update may be minor, affecting only a limited aspect of the LLM's functionality and leaving many prompts unaffected. An example of such unconfirmed alarms is shown in Figure 8, with more cases provided in Appendix D.3.

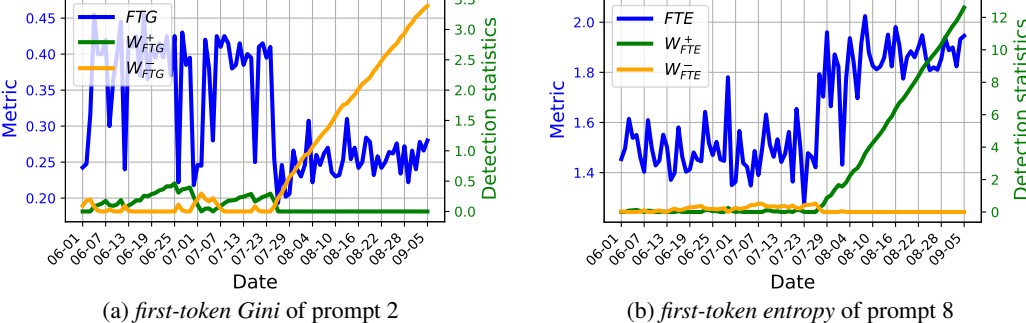

(a) *first-token Gini* of prompt 2    (b) *first-token entropy* of prompt 8

Figure 8: Illustration of unconfirmed changes detected for `gpt-4-turbo`. For several prompts in our set, the detection statistics show a significant increase beginning between July 23 and July 29.

## 5 CONCLUSION

In conclusion, our proposed online change detection method offers a computationally efficient solution for identifying changes in black-box LLMs. By leveraging a CUSUM-type detection statistic based on entropy and the Gini coefficient, combined with a UCB-based adaptive prompt selection strategy, our method quickly detects changes while controlling the false alarm rate. The evaluation results from both synthetic and real LLM API interactions highlight its effectiveness across various types of changes. This work offers a flexible framework and opens new opportunities for exploring the usage of alternative statistics beyond entropy and Gini, conducting further theoretical analyses on detection and selection performance, examining a wider range of change scenarios, and deploying this algorithm for continuous monitoring to ensure the integrity of LLM-powered applications.

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

## A   DERIVATION OF THE GINI COEFFICIENT FOR TOKEN DISTRIBUTION

The Gini coefficient quantifies inequality within a frequency distribution, such as income levels (Gini, 1921) and is traditionally used in economics. A Gini coefficient of zero represents perfect equality, where all individuals have identical income or wealth, while a Gini coefficient of one (or 100%) indicates maximum inequality, with all wealth concentrated in a single entity. It is defined as the ratio of the area between the Lorenz curve, which plots cumulative income against cumulative population, and the line of perfect equality, to the total area under the line of perfect equality. In the following, we derive the Gini coefficient on token probability distribution. See Figure 9 for demonstration.

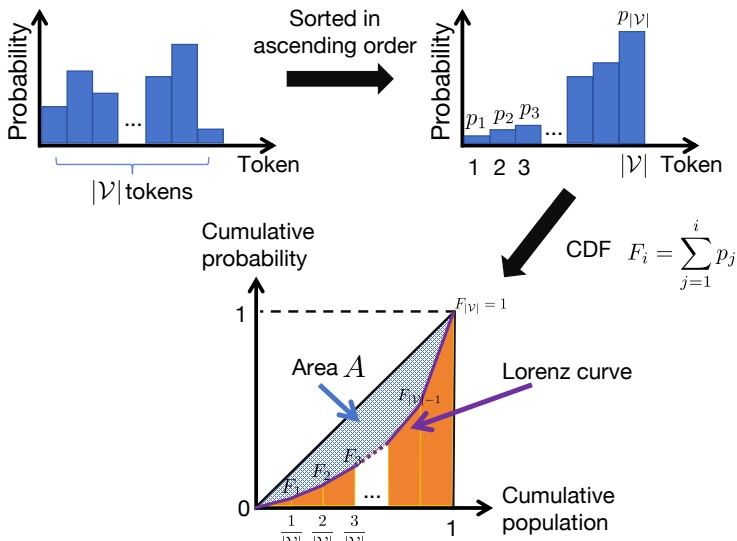

Figure 9: The computation of Gini coefficient on token probability distribution.

In our case, we take *first-token Gini* for example. We sort the probability distribution of tokens in vocabulary $\mathcal{V}$ in ascending order, with the $i$-th smallest probability being $p_i$. We accumulate the sorted probabilities to its cumulative distribution function (CDF) as

$$F_i = \sum_{j=1}^{i} p_j,$$

and we define $F_0 = 0$. The cumulative population refers to the proportion of popula- tion up to $i$-th token, and thus is $\frac{i}{|\mathcal{V}|}$ under our setting. We plot the curve with points $(0,0), (\frac{1}{|\mathcal{V}|}, F_1), (\frac{2}{|\mathcal{V}|}, F_2), \ldots, (1,1)$ in order, which is exactly the Lorenz curve. We denote the area under Lorenz curve as $A_0$. Then $A_0$ is computed as

$$A_0 = \sum_{i=1}^{|\mathcal{V}|} \frac{1}{2}(F_i + F_{i-1}) \cdot \frac{1}{|\mathcal{V}|}.$$

We further denote the area between the Lorenz curve and the line of perfect equality, i.e. the line segment connecting $(0,0)$ and $(1,1)$ as $A$. Then it is easy to get

$$A = \frac{1}{2} - A_0.$$

Since the total area under the line of perfect equality is $\frac{1}{2}$, according to the definition of Gini coeffi- cient, we can compute $\mathtt{FTG}(x)$ as $2A$, which is

$$\mathtt{FTG}(x) = 2A = 1 - \frac{1}{|\mathcal{V}|} \sum_{i=1}^{|\mathcal{V}|} (F_i + F_{i-1}).$$

Similarly, we can derive Gini coefficient for the joint distribution of the first $N$ tokens, which is $\mathtt{NTG}(x)$ in subsection 3.1.

## B    DETAILS ON SOFT WATERMARK

We review the following simplified soft watermark mechanism for next token generation in Kirchenbauer et al. (2023), parameterized by $\gamma$ and $\delta$. Here $\mathcal{V}$ denotes the vocabulary of an LLM.

1. Given an input prompt $x$, generate a logits vector $l \in \mathbb{R}^{|\mathcal{V}|}$ for the next token.

2. Randomly partition the vocabulary set into a green set and a red set, with the size of the green set being $\gamma|\mathcal{V}|$.

3. Apply a positive offset $\delta$ to the logits of the tokens belonging to the green set, i.e.,

$$\tilde{l} = l + \delta \cdot [\mathbb{1}\{\text{token}_1 \in \text{green set}\}, \dots, \mathbb{1}\{\text{token}_{|\mathcal{V}|} \in \text{green set}\}]^\top.$$

4. Pass $\tilde{l}$ to a Softmax operator to obtain probability vector $\hat{p}$ and sample the next token from $\hat{p}$.

The partitioning of the green set and the red set is determined by a watermark key. In practice, the key can be selected by the user, and its hash value serves as a random seed for the partitioning process, ensuring randomness in the division. We run the experiment using five random watermark keys, with the green list determined by each key and fixed once selected.

## C    EXPERIMENTS DETAILS

**LLM parameters setting**    In the synthetic change cases in Subsection 4.1, we set the model temperature to 1.0, sampling parameter top_p to 0.9 and no constraint on top_k. In the real world experiments, we set the LLM API's temperature parameter to 1.0 for Jamba and Cohere, and 1.5 for others. We still set sampling parameter top_p to 0.9 and no constraint on top_k.

**Prompts Used in Section 4**    The prompts used in section 4 are listed with index in Table 1. This prompt set is comprised of some curated prompts (with no prior knowledge to their potential sensitivity to changes), some of which are based on the idea of random choice generation (Tang et al., 2023) and some are math problems (Chen et al., 2024) or multiple choices. Although some prompts may appear similar, they generally elicit different responses from LLMs. The 20 prompts used in real-world cases (subsection 4.2) include all these prompts except for prompt 1, 4, and 5.

Table 1: List of all prompts used in Section 4.

| Index | Prompt |
|---|---|
| 0 | Generate 20 random numbers sampled from a normal distribution with a mean of 5 and standard deviation of 2 |
| 1 | What qualities do you look for in a mentor or leader? |
| 2 | Please give me a random number between 1 and 100: |
| 3 | Choose randomly one number from 1 to 100: |
| 4 | How does one develop creativity? |
| 5 | What do you think is the most important branch of mathematics for everyday life (arithmetic, algebra, statistics, geometry)? |
| 6 | Give me a random number in range 1 2 3 4 5 6 7 8 9: |
| 7 | Randomly select a letter in alphabet A-Z: |
| 8 | Given a list of fruits as Apple Banana Orange Grape Kiwi Pineapple Mango Strawberry Watermelon Lemon Lime Peach Pear Cherry Plum Avocado Coconut Raspberry Blueberry Blackberry Cranberry Papaya Fig Guava Lychee Dragonfruit Persimmon Passionfruit Pomegranate Cantaloupe, from the above list randomly choose one: |
| 9 | Tell me your favourite fruit in one word: |

| Index | Prompt |
|---|---|
| 10 | Choose your favourite color from red or green or orange? Your choice is: |
| 11 | Give me a random color from red green orange. Your word choice is: |
| 12 | The sum of 1 and 4 is: |
| 13 | Choose your favourite color from red or green or orange? Your choice will be: |
| 14 | Is 17077 a prime number? Think step by step and then answer "[Yes]" or "[No]". |
| 15 | Choose one random number from 1 to 100. Respond with only the number. |
| 16 | Randomly choose one fruit name. |
| 17 | You are taking a survey. Pick the best single option (e.g., A). Question: Please think about what things will be like in 2050, about 30 years from now. Thinking about the future of the United States, would you say you are: A. Very optimistic B. Somewhat optimistic C. Somewhat pessimistic D. Very pessimistic E. Refused Answer: |
| 18 | You are taking a survey. Pick the best single option (e.g., A). Question: Still thinking ahead 30 years, which do you think is more likely to happen in the U.S.? A. The U.S. economy will be stronger B. The U.S. economy will be weaker C. Refused Answer |
| 19 | You are taking a survey. Pick the best single option (e.g., A). Question: If you were deciding what the federal government should do to improve the quality of life for future generations, what priority would you give to reducing the gap between the rich and the poor? A. A top priority B. An important, but not a top priority C. A lower priority D. Should not be done E. Refused Answer: |
| 20 | Generate one random number between 1 and 100. For example, your response is 18 or 57. Remember that your response should only contain the number you choose. Then your response is: |
| 21 | Give me one random number from 1, 2, ... , 100 |
| 22 | Give me one random number from 0, 1, 2, 3, 4, 5, 6, 7, 8, 9: |
| 23 | Your response should only contain one number. Give me a random number from 1,2,3,4,5,6,7,8,9. |
| 24 | Provide a brief history of the Roman Empire and conclude with its influence on modern governance. |
| 25 | Recommend a book for someone interested in science fiction, but prefers a focus on character development. |

**Threshold Selection for Target ARL** In order to save computing effort in the determination of the thresholds under target ARL values (which is usually large), we adopt an efficient approximation algorithm that uses the fact that the distribution of stopping time $T$ under the pre-change regime is approximately exponential when ARL is large. Such approximate algorithms for determining $b$ have been widely adopted in online change detection; see Siegmund & Yakir (2008) for one example. Instead of simulating the mean of the distribution of $T := \inf\{t : W(t) \geq b\}$ directly, we obtained an estimate of the mean from an estimate of the cumulative distribution function of $T$ based on 20 iterations. Specifically, in each iteration, we simulate the pre-change trajectory with 100 time steps, and compute the maximum of the detection statistics at 100 time steps. These maximum values under 20 iterations are then denoted as $W_{1,\max}, W_{2,\max}, \ldots, W_{20,\max}$. For the desired ARL values $\Gamma = \mathbb{E}[T]$ where the expectation is taken under the pre-change regime, we approximate the stopping time $T$ as an exponential distribution with mean $\Gamma$. Thus we have $P(W_{\max} < b) = P(T > 100) \approx e^{-100/\Gamma}$. Thus the corresponding threshold $b$ can be approximated as the $e^{-100/\Gamma}$ quantile of the

set $\{W_{1,\max}, W_{2,\max}, \ldots, W_{20,\max}\}$. Note that we can also use more iterations and longer sequences within each iteration, which tends to improve the approximation accuracy.

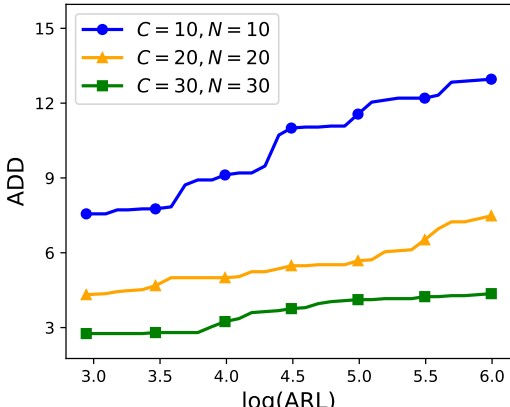

Figure 10: ADD v.s. ARL trade-off curves under different parameter settings.

**Choice of Parameter $C$ and $N$**   The choice of repeat times $C$ and token length $N$ concerns a trade-off between detection power and computation cost. With higher $C$, the estimation of our proposed four metrics becomes more accurate, thus lower We compare the Average Detection Delay (ADD) across different Average Run Length (ARL) levels under three parameter settings: 1) $C = 10, N = 10$; 2) $C = 20, N = 20$; 3) $C = 30, N = 30$, as shown in Figure 10. The results indicate that as $C$ and $N$ increase, the ADD decreases for a given ARL level. Notably, the ADD under $C = 20, N = 20$ is comparable to that of $C = 30, N = 30$, across ARL levels ranging from $e^{-6} \approx 0.2\%$ to $e^{-3} \approx 5\%$. However, the detection procedure with $C = 30, N = 30$ incurs nearly double the query cost. Therefore, we choose $C = 20, N = 20$ in our experiments for a better balance between detection performance and computational efficiency.

# D   MORE EXPERIMENTAL RESULTS

## D.1   MORE RESULTS FOR DETECTION WITH ONE PROMPT

**Trade-off Curve for Different Detection Statistics And different Watermark Strength**   For the watermark change detection with one prompt in subsection 4.1.1, we plot the trade-off curves between Average Detection Delay (ADD) and Average Run Length (ARL) in Figure 11. Details on the simulation of thresholds for different ARLs are provided in Appendix C. After obtaining the threshold under certain ARL, we repeatedly run the detection procedure for five times, and calculate the average detection delay (ADD). We also vary watermark strengths using the parameters $\delta$ and $\gamma$, where larger values of $\delta$ and $\gamma$ indicate stronger watermarks and more significant changes. Under each watermark strength, we only plot the trade-off curve for detection statistic $W$, which is the maximum of the four individual detection statistics. As shown in Figure 11a, our proposed detection statistic $W$ has a relatively small detection delay (more results can be found in Figure 12). This confirms the efficiency of our combined detection approach. From Figure 20b, we see that the detection delay increases as the watermark becomes weaker, with decreasing values of $\delta$ and $\gamma$.

**ADD-ARL Tradeoff for More Prompts**   In section 4.1.1, we state that generally different detection statistics will outperform in different settings, whereas the maximum of them, i.e. $W$ always maintains good performance. We illustrate this finding by prompt 12. Here we provide more evidence under other prompts in Figure 12.

**Detection Statistics Grow after Version Change: Demonstration for More Prompts**   Recall that we synthesize three version change cases in section 4.1.1. We show that our proposed detection statistics grow rapidly after the change point in all three cases using one prompt. Here we illustrate the detection statistics' detection power by showing the same kind of growing behaviour on more prompts. See Figure 13.

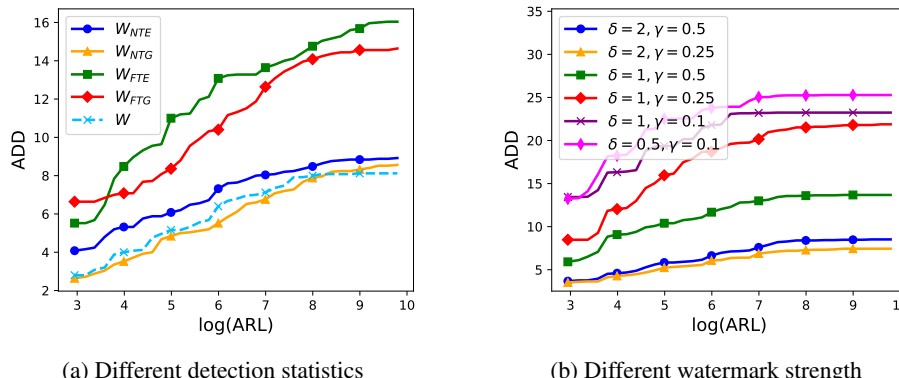

(a) Different detection statistics  (b) Different watermark strength

Figure 11: (a): Trade-off between ADD and ARL for different detection statistics. Our proposed detection statistics $W$, which is the maximum of the four detection statistics, achieves relatively small delays across all ARL levels. (b): Trade-off between ADD and ARL for different watermark strengths. As the watermark becomes weaker, the detection delay increases.

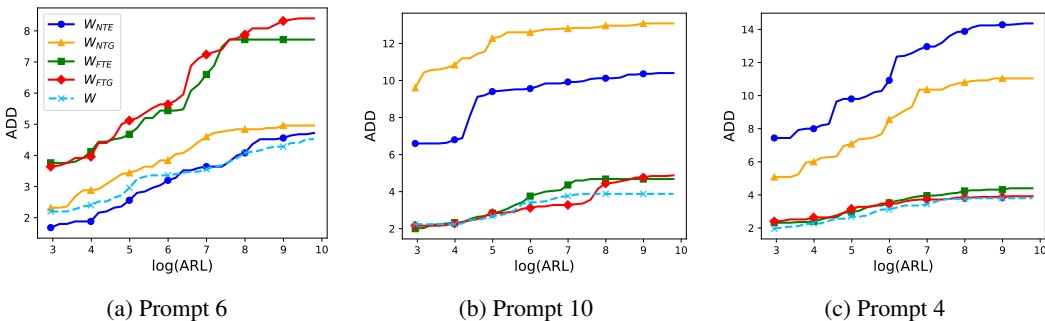

(a) Prompt 6  (b) Prompt 10  (c) Prompt 4

Figure 12: Trade-off curves of ADD and ARL for the four detection statistics and the maximum of $W_s$ under different prompts. We can easily find out that generally different detection statistics will outperform in different settings, whereas $W$ always maintains good performance.

### D.2 MORE RESULTS FOR DETECTION WITH ADAPTIVE SELECTION

**Detection with Adaptive Selection Converges to Prompts of High Sensitivity** From Figure 14 we also see the our proposed detection algorithm with adaptive prompts selection converges to prompts with the highest sensitivities, which are prompt 8, 9, 10, 12, and 13 under the current setting. Different runs may exhibit slight variations in the prompts to which the algorithm ultimately converges, but generally, sensitive prompts are selected quickly after the change happens.

### D.3 MORE RESULTS FOR DETECTION IN REAL-WORLD ONLINE DATA

**Confirmed Changes in Real-World APIs** Here we list more evidence that our detection algorithm captured the change in `mistral-instruct` at July 24th, 2024. See Figure 15.

**Unconfirmed Changes in Real-World APIs** We list two probable changes in real world LLM APIs which are not officially announced ot confirmed. The two changes are in `jamba-instruct` from AI21 Labs and `gpt-4-turbo` from OpenAI. We choose these two APIs because the detection statistics of many prompts and the corresponding four metrics experienced a surge almost simultaneously during a small interval of days. Thus we have comsiderably higher confidence to report them, as shown in Figure 16 and 17.

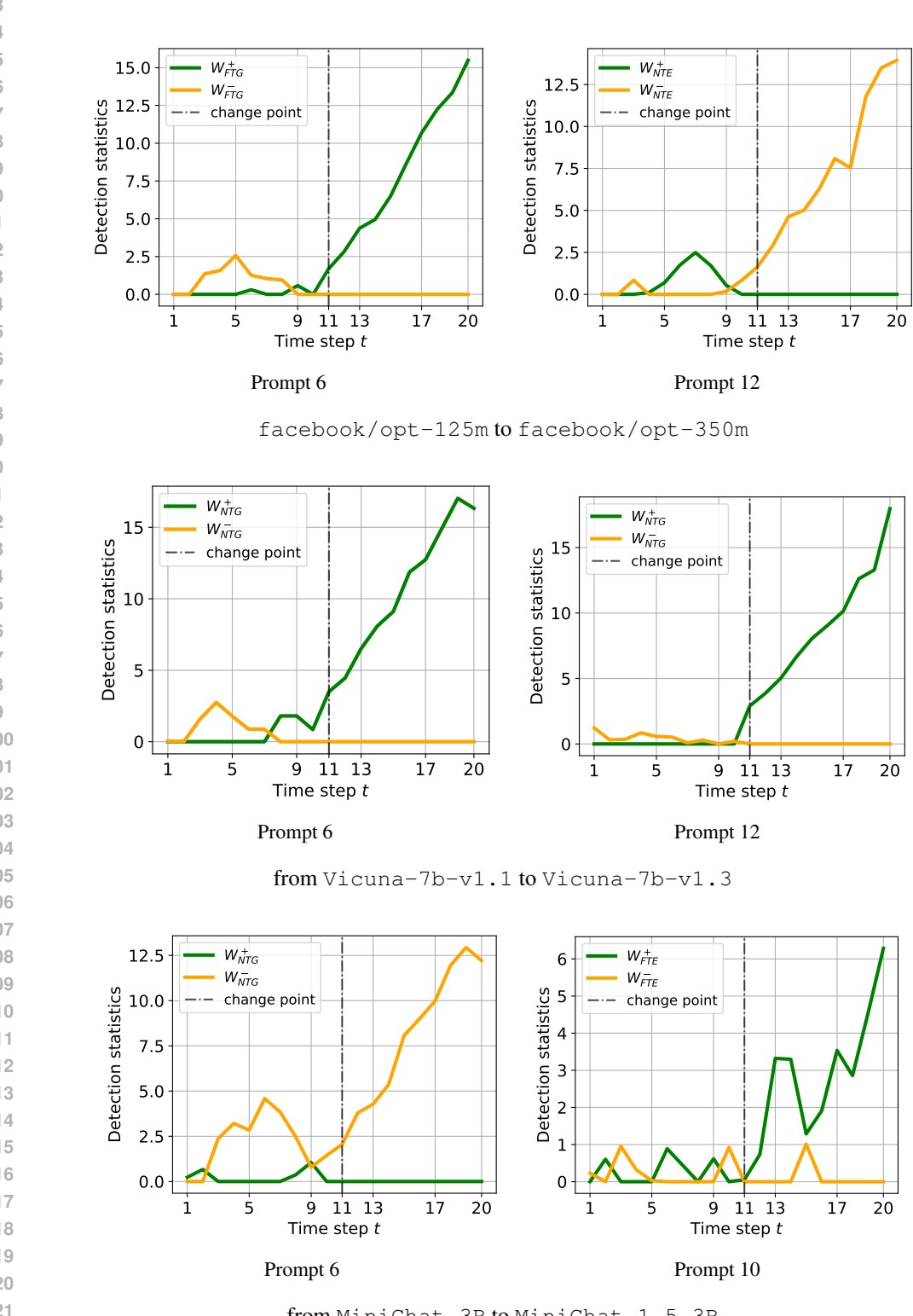

Figure 13: Demonstration for detection statistics growth after change point in version change cases.

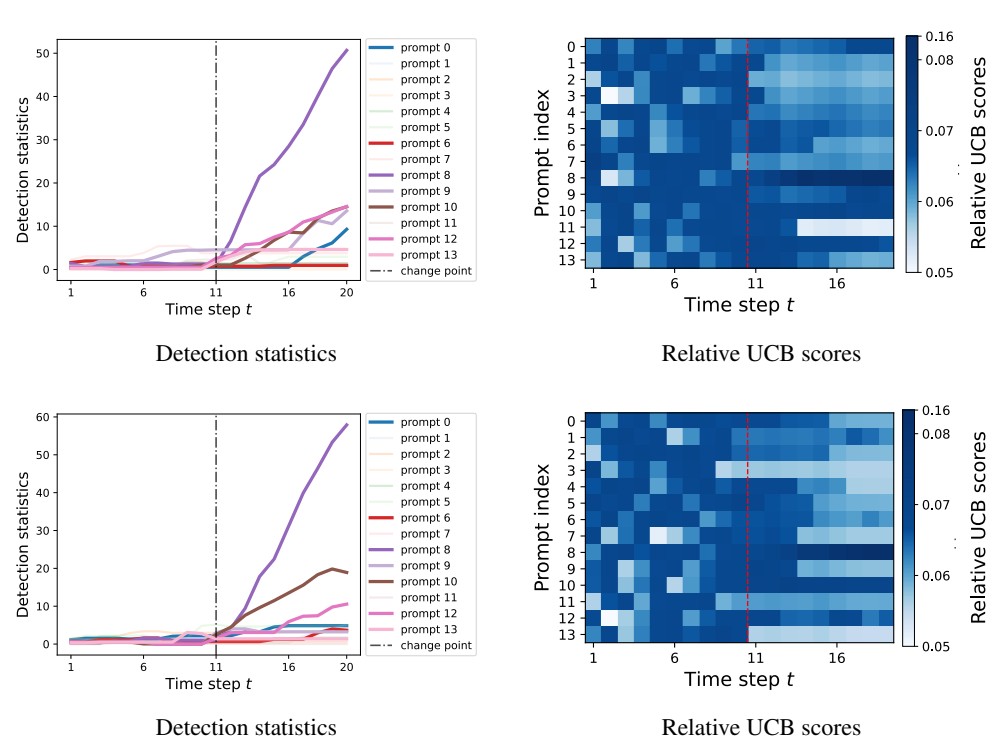

Figure 14: Repeated experiments for detection with adaptive selection.

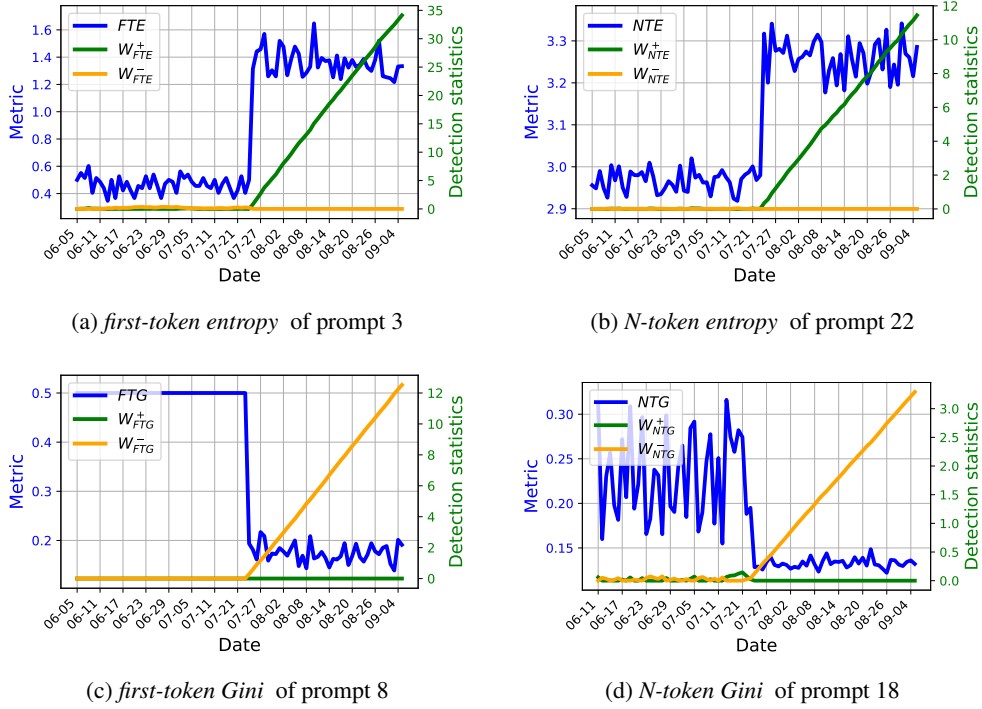

(a) *first-token entropy* of prompt 3

(b) *N-token entropy* of prompt 22

(c) *first-token Gini* of prompt 8

(d) *N-token Gini* of prompt 18

Figure 15: Confirmed change in `mistral-large-latest` on July 24th, 2024. We could see increasing detection statistics approximately between July 21st and 27th, 2024.

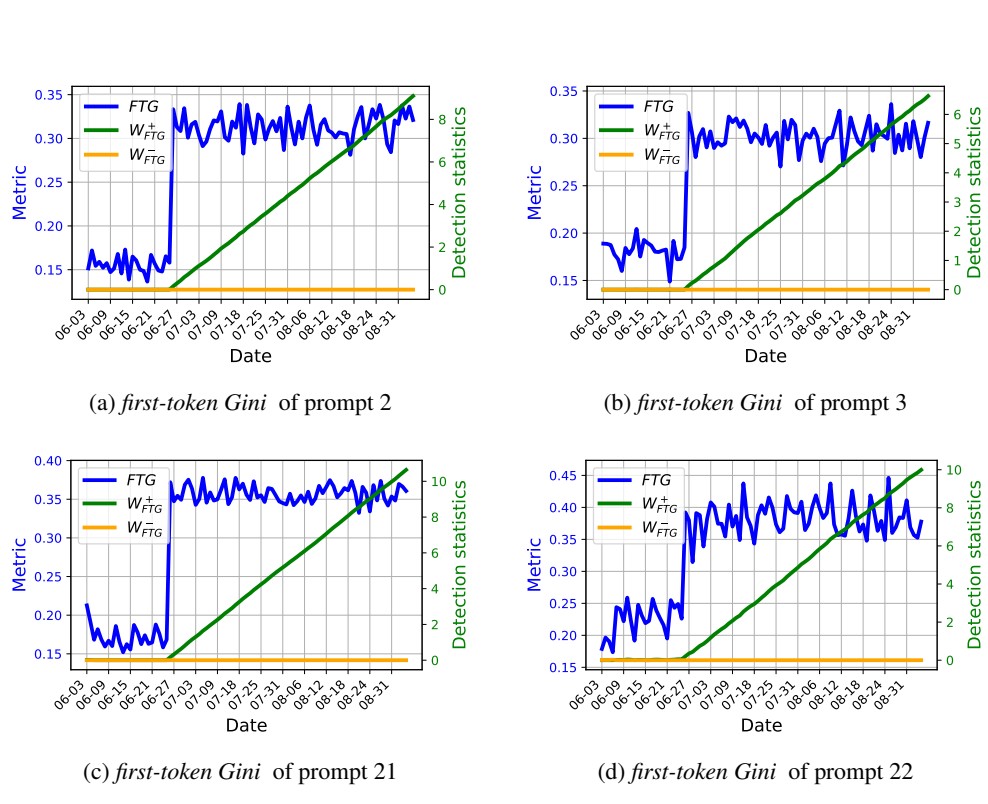

(a) *first-token Gini* of prompt 2

(b) *first-token Gini* of prompt 3

(c) *first-token Gini* of prompt 21

(d) *first-token Gini* of prompt 22

Figure 16: Unconfirmed change in `jamba-instruct`, approximately between June 21st and 27th, 2024. Here we use *first-token Gini* to illustrate, while other metrics behave similarly.

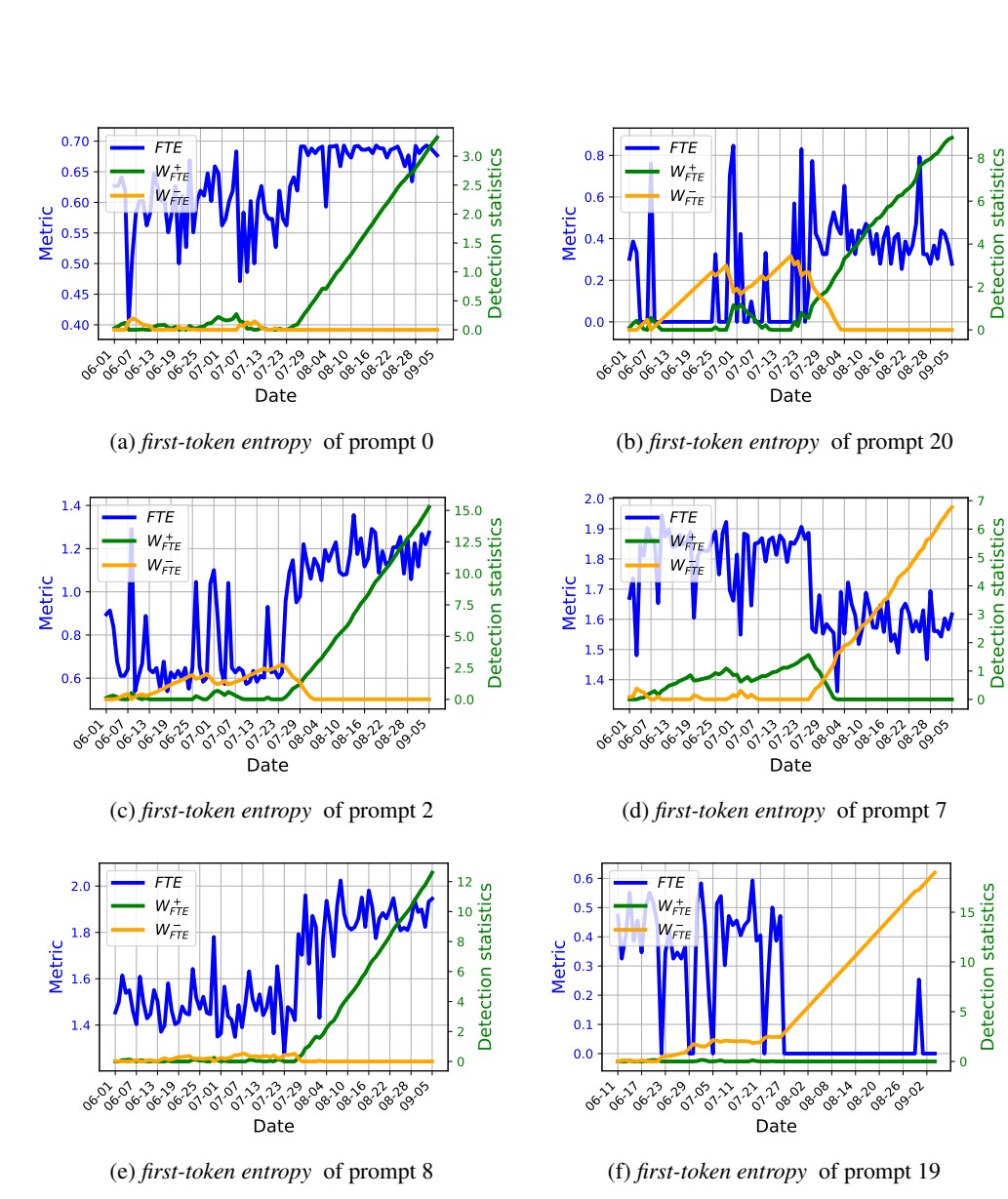

(a) *first-token entropy* of prompt 0

(b) *first-token entropy* of prompt 20

(c) *first-token entropy* of prompt 2

(d) *first-token entropy* of prompt 7

(e) *first-token entropy* of prompt 8

(f) *first-token entropy* of prompt 19

Figure 17: Unconfirmed change in `gpt-4-turbo`, approximately between July 23nd and 29th, 2024. Here we use *first-token entropy* to illustrate, while other metrics behave similarly.

# E    REBUTTAL

## E.1    DETECTION BASED ON TEXT SIMILARITY: A SIMPLE BASELINE

In this subsection, we consider a simple baseline based on text-level similarity for online change detection in LLM. This baseline works as follows. We again collect the responses for a given prompt $x$ for $C$ times at each time step $t$ during the detection procedure. We tokenize each response into a sequence of tokens and take the first $N$-tokens from each response at time $t$ to get a token set. Instead of calculating metrics on this token set as we did in our proposed detection algorithm, in the baseline, we convert this token set into a frequency vector $\mathbf{v}_t$, which captures the count of each token's occurrences within the set. We also convert the all historical responses for prompt $x$ into one token set, and get the historical frequency vector $\mathbf{v}_{\text{history}}$ using this token set. Then given a threshold $b \in (-1, 1)$, the detection procedure stops when the cosine similarity between $\mathbf{v}_t$ and $\mathbf{v}_{\text{history}}$ first drops below $b$, as

$$\frac{\mathbf{v}_{\text{history}} \cdot \mathbf{v}_t}{\|\mathbf{v}_{\text{history}}\| \|\mathbf{v}_t\|} \leq b.$$

In the following experiments, we keep the parameter configuration in the text similarity baseline identical to our proposed algorithm. Specifically, we set $C = 20, N = 20$, and we repeat the experiment for 5 times. We accessed both algorithms in two different scenarios: emergence of watermark and synthetic version change (from `facebook/opt-125m` to `facebook/opt-350m`). The results are shown in Figure 18 and Figure 19.

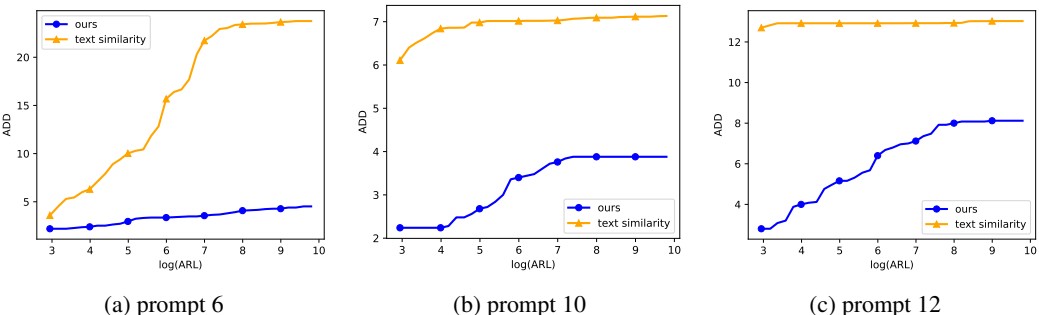

| (a) prompt 6 | (b) prompt 10 | (c) prompt 12 |

Figure 18: ADD and ARL trade-off comparison under the emergence of watermark. Under the same ARL, a lower ADD indicates a lower delay in average and thus a better performance. It is shown that our proposed detection algorithm outperforms the text similarity baseline.

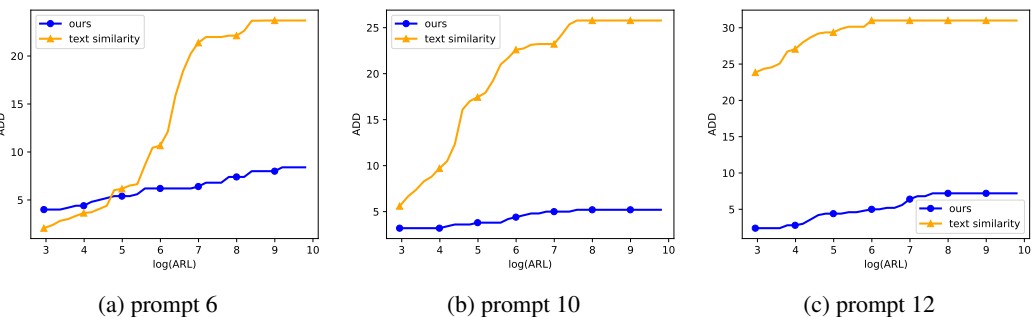

| (a) prompt 6 | (b) prompt 10 | (c) prompt 12 |

Figure 19: ADD and ARL trade-off comparison under synthetic version update from `facebook/opt-125m` to `facebook/opt-350m`. It is shown that our proposed detection algorithm outperforms the text similarity baseline in most false alarm rate constraints except for prompt 6 at small ARL.

It is shown that when ARL is large (meaning a low tolerance for false alarms), the baseline method suffers from a surge in delay, but our detection algorithm still performs well. Such disparity may be attributed to the cumulative nature of our algorithm. To illustrate this reasoning, we plot the trajectory with time for both our detection statistics and the text similarity baseline. For a fair

comparison, when a target ARL is assigned, we obtain the two corresponding thresholds for both methods, and make sure that the two thresholds lead to the same ARL value. Then under the same setting where the change scenario is the emergence of watermark, the prompt used is prompt 6, $ARL = 10,000$ and $\nu = 11$, we run both algorithms and record their statistics evolution with time, as shown in Figure 20. It can be seen that given the same level of false alarm rate, the text similarity is less likely to hit the threshold due to two possible reasons: 1) Text similarity does not enjoy a low variance property, which leads to a lowered threshold to compensate for pre-change instability. This lower threshold makes it harder for post-change text similarity to reach the threshold. 2) The non-cumulative nature of this baseline method prevents it from accumulating deviations from normal values, thus exacerbating the problem mentioned in the first reason.

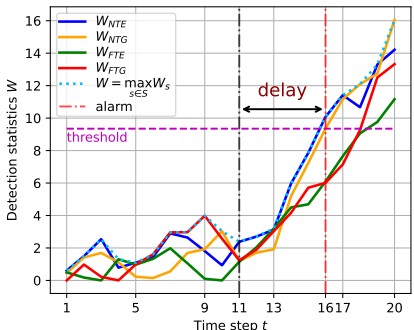

(a) Our cumulative type detection method

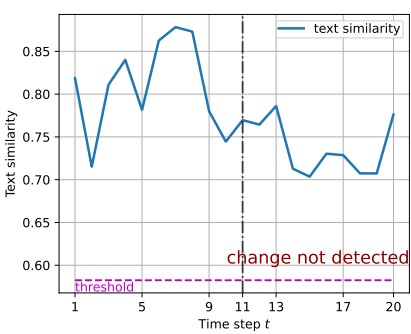

(b) Text similarity based detection method

Figure 20: (a): The detection statistics in our algorithm under the above specified setting. Our detection method successfully detected a change. (b) Text similarity in the baseline method. The threshold is set corresponding to the same level of false alarm rate as (a). The baseline method fails to detect the change in this case.

### E.2 DEMONSTRATION FOR RESPONSES COLLECTED IN PRE-CHANGE STAGE AND POST-CHANGE STAGE

To demonstrate that the responses change in an inconspicuous way, we present some responses generated by pre-change LLM and post-change LLM for comparison. See Table 2. In this table, group 1 refers to the change from `opt-125m` to `opt-125m` watermarked with key 0, using prompt 10; group 2 refers to the change from `vicuna-7b-v1.1` to `vicuna-7b-v1.3`, using prompt 12; group 3 refers to the real-world change in Mistral API, where change happened at July 24, 2024 and pre-change responses are selected from July 07 and post-change responses are from July 25, using prompt 0; group 4 refers to the change from GPT-4o to GPT-4o with prompt injection specified in Subsection E.3 using prompt 23, and 20 responses are provided.

Table 2: Responses comparison between pre-change and post-change LLM.

| Group | Pre-change responses | Post-change responses |
|---|---|---|

| Group | Pre-change responses | Post-change responses |
|---|---|---|
| 1 | (1) Green, orange, or red. Your choice is: blue, green or orange, or red...
(2) Red, green, orange, or black. To choose your favourite colour, click here to place it.
(3) Red or green. If you want your items to ship before holiday, check with your local customs
(4) Or black or blue, Alternatively yellow or yellow, Alternatively white or blue, or any.
(5) Cyan and pink are the best colours for LGBT people in our society. | (1)Red or green, your choice is: orange, your choice is: blue...
(2)red, yellow or green, orange or orange. Choose from a selection of eight colour combinations. Each
(3) red, green or orange. If this color is already in your wardrobe, you may want to check"
(4) Blue, green, or red. Add one final touch, a mysterious alphabet. The easiest choice is
(5) Pink, yellow, silver, or brown. These are just a few of thousands of choices to pick. |
| 2 | (1) 5, so the expression "1 + 4" is equivalent to "5". Qed
(2) 5. Let me know what you think of this problem. "Well, it's a pretty...
(3) 5. The sum of 2 and 3 is 5. The sum of 4 and...
(4) 5, so it follows that 2 and 3 also make 5. Suppose...
(5) 5, which is a multiple of 3. The sum of 2 and 3 is... | (1) 5. This is true, as 1 + 4 = 5.
(2) 5. So you have to be extra careful with adding numbers in pairs, so it doesn't
(3) 5. The sum of 2 and 5 is 7. The sum of...
(4) 5, so we can start at either 1 or 4. Let's Xavi:
(5) 9. The product of 3 and 4 is 12. The quotient |
| 3 | (1) Sure, here are 20 random numbers sampled from a normal distribution...
(2) Here are 20 random numbers sampled from a normal distribution...
(3) Sure, I can help with that. Here are 20 random numbers...
(4) Sure, I can generate those for you. Here are 20 random numbers sampled from a...
(5) Sure, I can generate those for you. However, as a text-based AI, I can... | (1) Sure, here is a list of 20 random numbers sampled from a...
(2) Sure! Here are 20 random numbers sampled from a normal distribution...
(3) Sure, I can help generate a list of 20 random numbers sampled from a normal...
(4) To generate 20 random numbers sampled from a normal distribution with a mean...
(5) Certainly! Here are 20 random numbers sampled from a normal distribution ... |
| 4 | 4,4,3,5,4,4,6,5,7,7,4,7,6,6,7,6,5,7,3,5 | 7,5,7,7,7,4,4,3,7,7,7,7,5,4,4,4,7,5,3,5 |

### E.3 Robustness of the Detection Algorithm

In our standard setting, given a prompt $x$, the responses distributions of both pre-change LLM and post-change LLM are assumed to remain unchanged within their respective regimes. This assumption holds when the LLM's context configurations, such as temperature and system message, remain consistent across all the interactions at different time points. To the best of our knowledge, this condition is typically met in most real-world user-LLM interactions unless explicitly modified by the user.

Yet, we demonstrate the robustness of our detection algorithm: even with slight perturbations in the pre-change and post-change response distributions, the algorithm can still detect changes quickly under a given false alarm rate constraint. To simulate the slight perturbation in response distribution, at each time step, we set the temperature of the LLM as a random variable uniformly sampled from the interval $[0.9, 1.0]$. We assess our detection algorithm's performance under this setting in the case

of emergence of watermark. All other parameter configuration stay consistent to Subsection 4.1.1. The results are shown in Figure 21. It is demonstrated that our detection algorithm is robust to the slight perturbation in response distribution when the false alarm rate constraint is relatively loose (meaning a small ARL). Yet the detection power degrades notably when the the false alarm rate constraint is relatively high.

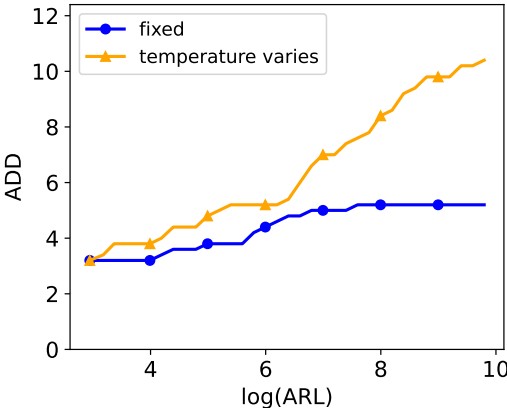

Figure 21: ADD v.s. ARL trade-off curve for different LLM generation setting during simulation. Here, "fixed" refers to the setting that LLM's temperature is fixed at 1.0 during generation, and "temperature varies" refers to that at each time step, LLM's temperature is uniformly sampled from $[0.9, 1.0]$.

### E.4 RESULTS FOR EMERGENCE OF PROMPT INJECTION

Prompt injection is a technique used to manipulate the behavior of language models by inserting specific instructions or prompts, often aiming to override or alter the original prompt's intended output. This method has gained attention due to its implications for security and reliability in LLM deployments. In our experiments, we also use prompt injection as an instance: starting from the change point $\nu$, the system message of the LLM is changed. This scenario can occur when an untrustworthy chatbot website is vulnerable to prompt injection, or when multiple users share a single chatbot account, allowing one user to exploit the web application's memory to influence the LLM's behavior. In this subsection, we compose an instance of prompt injection using GPT-4o API, and implement preompt injection through the system message of the API. We still use the same setting as specified in Subsection 4.1.1. We set the pre-change system message as

```
"You are an assistant designed to help users."
```

and the post-change system message as

```
"You're a helpful assistant.  However, you always
neglect the last five words in the prompt."
```

After running our detection algorithm in this case, we obtain results in Figure 22. It is clearly presented that our detection statistics still have the detection power to the emergence of prompt injection, highlighting its potential application in LLM security and reliability.

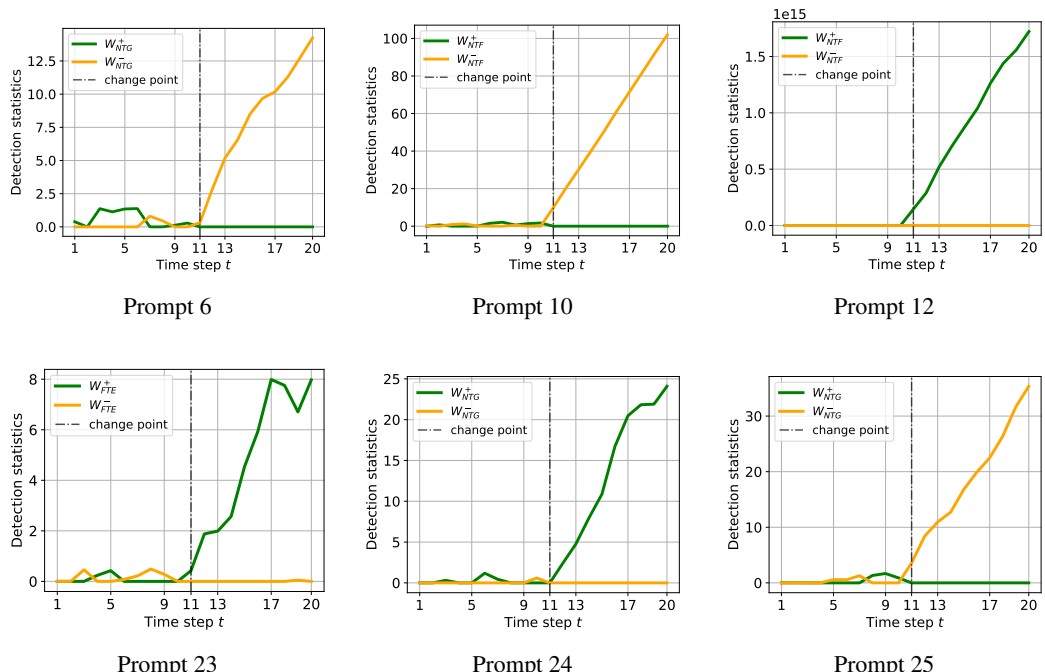

Figure 22: Demonstration for detection statistics growth after change point in prompt injection cases.

