# OpenReview forum: "Online Detection for Black-Box Large Language Models with Adaptive Prompt Selection"
_ICLR.cc/2025/Conference — Submitted to ICLR 2025_

### Official Review · Reviewer_EvAs · 2024-10-26

**Soundness:** 3
**Presentation:** 2
**Contribution:** 3
**Rating:** 6
**Confidence:** 3

**Summary:**

This work proposes a detection algorithm for online black-box LLM changes. The detection leverages a CUSUM-type statistic that incorporates entropy and the Gini coefficient. To enable effective detection, it adopts a UCB-based adaptive prompt selection strategy to identify the most change-sensitive prompts. In evaluations, the algorithm demonstrates effectiveness in identifying various backbone changes and model watermarking.

**Strengths:**

- The paper addresses the important issue of detecting online changes in black-box LLMs.
- The proposed approach is practical and applicable across different type of models, as it comes with minimal assumptions of model itself.
- The UCB-based adaptive prompt selection enables cost-effective detection by focusing on the most sensitive prompts.
- The method is robust, maintaining good ADD with a low false positive rate (high ARL).

**Weaknesses:**

- While the authors discuss hyper-parameter selection, further ablation studies on individual hyper-parameters (e.g., C and N) would help clarify their specific effects, rather than assessing them as a whole.
- Due to computational limitations, most experiments are conducted on relatively small models. An analysis of larger models (e.g., Vicuna-7B) would provide more insights into real practice. For example, how ARL and ADD will change with larger model sizes? Will this method still be robust with larger models?
- There are limited experiments on real-world APIs with LLM updates. Additional results on version changes will be helpful. For example, the author can test different versions of gpt-4o or gpt-4 with different dates from OpenAI API.

**Questions:**

- Could you clarify the calculation methods for NTE and NTG? Specifically, how are the tokens merged to estimate the distribution?
- Will different decoding strategies affect the robustness of the detection? For instance, is the model change less likely to be detected with nucleus sampling than with greedy decoding?

---

> ### Author Response · Authors · 2024-11-20
>
> Dear reviewer EvAs,
>
> We thank the reviewer for the comments, and we appreciate the time you spent on the paper. Below we address the concerns and comments that you have provided.
>
> > **Q1**: Clarification on the calculation for NTE and NTG
>
> **A1**: Due to the exponential growth of the total combinations of N tokens with regard to N, we designed an approximation method (see Section 3.2 in our paper). At each time step, we collect C responses for prompt x, each with length N. After tokenizing these C responses, we will obtain at most C*N tokens. We merge all of these C*N tokens into a set and count the frequency of each token in this set to get an empirical distribution. Then both NTE and NTG metrics are computed using this empirical distribution acquired from the C*N tokens.
>
> > **Q2**: Regarding different decoding strategies
>
>  **A2**: Different decoding strategies may affect the distribution of the generated response, then for two different models, the detection task will be easier if the detection strategy is adopted such that difference in the resulting pre-change and post-change response distributions are maximized. However, we would like to emphasize again that the decoding strategy is not the focus of our work and we treat the decoding strategy as a pre-determined component of the black-box LLM models.
>
> > **Q3**: Regarding the performance on larger LLMs
>
>  **A3**: We assessed the performance of our detection algorithm in relatively large LLM, such as version update from  vicuna-7b-v1.1 to vicuna-7b-v1.3 (see **Figure 4b**),  from MiniChat-3B to MiniChat-1.5-3B (see **Figure 4c**), and all real-world APIs (the parameter size of the LLMs behind them is often in trillions, see **Subsection 4.2**). Moreover, we **added a new experiment** under the case of prompt injection and used GPT-4o API for an instance (see Appendix E.4). All these experiments are conducted with large-size LLMs, and the effectiveness and robustness of our detection algorithm are validated.
>
>
> > **Q4**: Regarding more real-world changes
>
>  **A4**: We collected data of 9 various LLM APIs, as specified in Section 4.2 in our paper (they are: gpt-4o, gpt-4, gpt-4-turbo, gpt-3.5-turbo, Cohere command-r-plus, claude-3-haiku-20240307, mistral-large-latest, jamba-instruct and j2-ultra). We collected data from these nine APIs on a daily basis from June 01 to August 31. With the collected data, we successfully detected one confirmed change and two unconfirmed changes using our detection algorithm. With regard to gpt-4 and gpt-4o that you have mentioned, we indeed did the same study but there is no evidence for a change happening during this period. Yet the success of our algorithm on other real world cases have proved the effectiveness of our detection framework.
>
> > **Q5**: Regarding the choice of parameter C and N
>
>  **A5**: We would like to clarify that C and N is a relatively flexible hyperparameter during real world usages, and the key to choose C and N configuration is to balance the performance of the algorithm under this config and the cost of the algorithm, as we mentioned in Appendix C paragraph 3. It is easy to expect that the increase in C and N will enhance the algorithm's detection power. As a result, in real world usage, what we need to do is to set a larger C and N within our affordable computation cost. And for the effectiveness evaluation of the detection algorithm in our research, we just choose a hyperparameter configuration that almost reveals real performance of our algorithm and meanwhile can be affordable to most users so that they could also use it.
>
> Thank you again for your comments on our paper, which have guided us toward a more refined revision. We sincerely hope that we have addressed your concerns. If we have misunderstood your point, we welcome any further clarification of your concerns and hope that you may consider revising your evaluation of our work.

---

> > ### Comment · Reviewer_EvAs · 2024-11-24
> >
> > Thank you for your response! Some of my questions are solved, and I will maintain my score.

---

### Official Review · Reviewer_wJQu · 2024-10-27

**Soundness:** 2
**Presentation:** 1
**Contribution:** 2
**Rating:** 6
**Confidence:** 3

**Summary:**

This paper offers a CUSUM statistical method to detect if there is any change in underlying language models based on prompts. Towards this, it introduces a statistic based on entropy and Gini coefficient. The different methods introduced in the paper are effective in black-box settings and adaptive. The paper also discusses the detection of watermarking being enabled by model providers.

**Strengths:**

- The method is running aggregation of simple statistical metrics and is easy to compute.
- Strong empirical results for such simple metrics in detecting a change in a given model with little delay.
- The experiment covers both white-box and black-box API models.

**Weaknesses:**

- The prerequisites of the detector are unclear. The assumptions and level of access/information are missing and therefore the precise motivation/utility behind detection is unclear. The change in model is well advertised in "real-world APIs" and is publicly available information.
- If the detector has access to log probabilities from the model API, then it can easily hash them for each prompt and observe changes in them. The tok-K log probabilities are generally available for frontier SOTA models like GPT4, Gemini, and Claude. This simple baseline (for the detection of change in the model, not watermarking explicitly) is missing.
- The experiments only cover single instances of changes in the model. It would be great to see the True Positive Rate @ low False Positive/Alarm Rate aggregated over multiple different changes in API models. This can be difficult in black-box API case but can be easily done in a simulated case.
- For experiments, there are no baselines identified/used making it hard to evaluate effectiveness of method and the task. The related work section is poor.

**Questions:**

1. What is the threat model or detector's operation surface here? Does the detector have access to API and thus to log probabilities produced by it?

---

> ### Author Response · Authors · 2024-11-20
>
> Dear reviewer wJQu,
>
> We thank the reviewer for the comments, and we appreciate the time you spent on the paper. Below we address the concerns and comments that you have provided.
>
> > **Q1**: Regarding the level of information and model access
>
> **A1**: We focus on the black-box setting where log probabilities **cannot** be obtained. In fact, the top-K probabilities are not available in many cases, such as Mistral, Cohere, Claude covered in our experiments, and o1-preview most recently. Thus we focus on the black-box and generalized scenario, where the log probability is not available, or even if it is available, the user may not necessarily interact with the LLM via an API and thus only the response is available for change detection.
>
> > **Q2**: Regarding the concern that model update is publicly available information
>
> **A2**: LLM API providers often release their new version of API with a public announcement. However, this is not always the case. In fact ,many “minor” updates are released without public notification. We reference [this paper](https://arxiv.org/pdf/2307.09009) to support the relevance of detecting changes even in the absence of unauthorized parameter updates. In this paper, the authors showed an interesting fact that even in an LLM API with no official notification of version update, its performance can still vary across different times. And in our paper, we also detected changes in Jamba and GPT-4-turbo respectively, despite there being no announcement on their provider’s website. We believe this discovery underscores the value of our work.
>
> > **Q3**: Regarding the call for more change instances
>
> **A3**: We have provided two synthetic cases and one real-world case in our paper, and each of these cases include multiple instances (such as different model versions, different watermark keys etc.). For further evaluation, we now have also added a case of prompt injection, on which our detection algorithm also has strong performance (see Appendix E.4 in the revised draft). All of the experiments under these settings are conducted with at least 3 prompts and repeated for 5 times. We also provided the delay-ARL analysis (hope this is what you mean by “True Positive Rate @ False Positive Rate”) for the case of watermark (in Figure 11) and the case of version change from opt-125m to opt-350m (in Figure 6b). For other cases where we didn’t provide the delay-ARL analysis, we provided their trajectory of detection statistics. In these trajectories, we see a surge in detection statistics after the change happens.
>
> > **Q4**: Regarding baseline
>
> **A4**: We have implemented the “text similarity” baseline. The idea is that we convert the texts at each time step and also of history to a frequency vector, and compute the cosine similarity of the frequency vector at each time step and that of the history. When the similarity is below a threshold (selected to meet some false alarm rate constraint), the detection algorithm raises an alarm. We present the trajectory of different detection statistics on the same sequence in the **newly added Figure 20**, and in Appendix E.1 of the revised version, you will find that our proposed method has a much shorter delay (detect more quickly).
> We also perform the experiment multiple times and plot the average delay v.s. ARL (false alarm rate metric).  All results can be found in the **newly added Figure 19**. You will see that in most cases, our cumulative type of detection statistics outperform the text similarity baseline.
>
> Thank you once again for your comments on our paper, which have helped guide us toward a more thorough revision. We sincerely hope that we have addressed your concerns and that you may consider revising your evaluation of our work.

---

> > ### Author Response · Authors · 2024-11-25
> > **Follow-Up on Reviewer Feedback**
> >
> > Dear Reviewer wJQu,
> >
> > We hope this message finds you well. We are following up regarding our paper during the rebuttal phase. We have carefully addressed your concerns and made improvements to the paper accordingly.
> >
> > Since we are approaching the end of the discussion period, we kindly ask if your concerns have been adequately addressed by our rebuttal and revision. If so, we would greatly appreciate it if you could consider updating your evaluation. If there are any remaining concerns, we are happy to follow-up.
> >
> > Thank you for your time and consideration.
> >
> > Best regards,
> > Authors 13317

---

> > > ### Comment · Reviewer_wJQu · 2024-11-29
> > >
> > > Thanks for the author's rebuttal. I appreciate adding the baseline. I would have preferred to see a stronger baseline such as a simple bloom filter on completions and probability scores (that's possible with the current standard of returning top-5 logprobs) over some choice of datasets. My main issue, however, (not responded to directly in rebuttal text) is that there is no concrete and _realistic_ threat model. But this is more of an _opinion_ that this problem is not _realistic_ from either the attacker's or the defender's perspective -- so I will update the score to 6 and let the meta-process consider my thoughts for the final decision.

---

### Official Review · Reviewer_TL1q · 2024-10-28

**Soundness:** 1
**Presentation:** 2
**Contribution:** 1
**Rating:** 3
**Confidence:** 5

**Summary:**

The paper proposes an online changepoint detection method aimed at identifying changes in black-box LLM APIs. The method is evaluated on synthetic data through simulated watermarking and version updates

**Strengths:**

+ The paper is easy to follow
+ The author tries to propose a new topic related to LLM security

**Weaknesses:**

First of all, I found the study itself a little bit weird. The author claims that the detection is for "tampering, malicious prompt injection, or unauthorized parameter updates" at the very beginning of the paper. However, as the user of the API(since it is the black-box setting, I can only assume that the user wants to detect the API change since it makes no sense that the LLM developer wants to detect LLM change in the black-box setting), I don't find it very convincing to claim the ''unauthorized'' parameter updates as a concern that we even need a detector for that. The LLM provider has the right to update their model, and those trusted providers like OpenAI and Anthropic often inform users of the updates and have checkpoints for older versions. Also, the watermark or fingerprint is a method to protect the model or response property [1] , leaking them would make it easier for intellectual leakage. And the decoding changes should not be necessary connected with the watermark. For example, having a more efficient decoding method like [3], or decoding-phase alignment[4] could also lead to that. There is not necessary proof that it is harmful and I don't understand why it is a concern that the user must have the detection against it. For those untrusted LLM API/interface providers, then it could be a risk by prompt injection. For example, custom GPT store or Coze have lots of uncensored bots that the system prompt can be changed into malicious and leading to security or privacy risks[2]. From my side, I can only agree that this could be a risk that we should take care of. However, even though the author mentions the prompt injection in the introduction and abstract, there's no further study on that. Please clarify and provide specific scenarios where detecting such changes is necessary.

Second, I don't find the detection in this setting very challenging. Since some experiment details are missing, I can only assume that the API usage is with a fixed system prompt and zero temperature. In such case, the response is expected to be deterministic. If I see the changes as shown in Fig1, I should know obviously that the API might change. I don't think there's a need for such algorithm proposed in the paper. Could you clarify the experimental setting and provide the LLM response between different LLM versions? Also, have you considered some simple methods, such as similarity in the text level? The lack of baseline methods also makes the contribution less significant.

For the false positive measurement,  I believe a more accurate method is to evaluate the same model with different temperatures or different system prompts since this is more practical in the real-world. For example, the ChaGPT web interface's system prompt and temperature are unknown to the user(let' put aside why this is a risk). Using generating a different answer can sometimes have a huge difference in the response. It is a more practical case and the false positives should be take care with if we really need a detector here. How would your detector perform in this scenario?

Minor weakness: no board impact nor limitation discussed in the paper, also, how the author generated those prompts in the table are unknown

[1] Instructional Fingerprinting of Large Language Models

[2] On the (In)Security of LLM App Stores

[3] Fast Inference from Transformers via Speculative Decoding

[4] Decoding-time Realignment of Language Models

**Questions:**

See the weakness section above

---

> ### Author Response · Authors · 2024-11-20
>
> Dear reviewer TL1q,
>
> We thank the reviewer for the comments, and we appreciate the time you spent on the paper. Below we address the concerns and comments that you have provided.
>
> > **Q1**: Regarding the detection setting
>
> **A1**: Thank you for your detailed comments on various API change scenarios. Indeed, our goal is to detect model changes from the user's perspective when interacting with a black-box LLM. And we would like to clarify that our primary focus is on designing an effective and computationally efficient online detection framework that users can flexibly apply to detect changes, rather than identifying specific types of changes or diagnosing their root causes. By developing an algorithm capable of raising alerts when any model change is detected, we aim to provide users with timely insights that may assist them in relying on LLM responses for certain tasks. While root cause diagnosis (determining the type of change) could be an interesting direction for future research, it is complex even in white-box or traditional scenarios, and especially so in black-box LLMs with countless potential causes. For instance, while we use watermarking to simulate a decoding change, our aim is not to diagnose its cause after a successful detection. Additionally, we reference [this paper](https://arxiv.org/pdf/2307.09009) to support the relevance of detecting changes even in the absence of unauthorized parameter updates. In this paper, the authors showed an interesting fact that even in an LLM API with no official notification of version update, its performance can still vary across different times.
>
>
> > **Q2**: Regarding prompt injection
>
> **A2**: We added new experiments on prompt injection. We conduct the prompt injection through changing the system message of GPT-4o API, and we validated the effectiveness of our algorithm in this scenario. For details, please see the **newly added Figure 22 and Appendix E.4**.
>
> > **Q3**: Regarding the necessity of online detection algorithms
>
> **A3**: We would like to clarify that the example shown in Fig. 1 is purely illustrative and does not represent the actual responses we are working with. In fact, developing online change detection methods is a long-standing challenge, as many distributional changes cannot be detected by examining individual data instances alone. The simplest example is that data observations from N(0,1) and N(0.01,1) are sufficiently close and cannot be distinguished by individual points, yet cumulative sum type detection methods can effectively identify such changes quickly. Similarly, in our LLM detection setting, below we provide a side-by-side comparison of the **real** responses collected in our experiments, generated by different LLM models:
>
> Prompt: The sum of 1 and 4 is:
> | Pre-change Model                                         | Post-change Model                                      |
> |----------------------------------------------------------|--------------------------------------------------------|
> | 5, so the expression ”1 + 4” is equivalent to ”5”. Qed   | 5. This is true, as 1 + 4 = 5.                         |
> | 5. Let me know what you think of this problem. ”Well, it’s a pretty... | 5. So you have to be extra careful with adding numbers in pairs, so it doesn’t |
> | 5. The sum of 2 and 3 is 5. The sum of 4 and...          | 5. The sum of 2 and 5 is 7. The sum of...              |
> | 5, so it follows that 2 and 3 also make 5. Suppose...    | 5, so we can start at either 1 or 4. Let’s Xavi:       |
> | 5, which is a multiple of 3. The sum of 2 and 3 is...    | 9. The product of 3 and 4 is 12. The quotient          |
>
>
>
>
> As shown in the example above, the API change is not immediately apparent from reading the responses, as they appear quite similar. More examples like this are provided in **Appendix E.2**. This underscores the necessity of developing online detection methods tailored to LLM models.
>
> > **Q4**: Clarification on experimental settings
>
> **A4**:  Temperature is non-zero throughout all our experiments. In synthetic cases, we set temperature to default value 1.0, and in real-world APIs we set temperature to 1.5 for most APIs and 1.0 for Jamba and Cohere.  We have added clarifications on experimental details in Appendix C paragraph 1.
>
> **To be continued ...**

---

> > ### Author Response · Authors · 2024-11-20
> >
> > **Cont.**
> >
> > > **Q5**: Comparison with the text similarity baseline
> >
> > **A5**:  We have implemented the “text similarity” baseline. The idea is that we convert the texts at each time step and also of history to a frequency vector, and compute the cosine similarity of the frequency vector at each time step and that of the history. When the similarity is below a threshold (selected to meet some false alarm rate constraint), the detection algorithm raises an alarm. We present the trajectory of different detection statistics on the same sequence in the **newly added Figure 20**, and in Appendix E.1 of the revised version, you will find that our proposed method has a much shorter delay (detect more quickly).
> >
> > We also perform the experiment multiple times and plot the average delay v.s. ARL (false alarm rate metric).  All results can be found in the **newly added Figure 19**. You will see that in most cases, our cumulative type of detection statistics outperform the text similarity baseline.
> >
> >
> > > **Q6**: Regarding varying temperatures
> >
> > **A6**:  We have added a new experiment where the temperature of both pre-change and post-change LLM are allowed to be uniformly sampled from interval [0.9,1.0]. We assess the performance of our detection algorithm under this case using the watermark emergence. In the **newly added Figure 21** in Appendix E.3, you can see that the performance of the detection algorithm does not suffer from an obvious degradation when log(ARL) is below 6, which satisfies the need for false alarm rate in daily usage.
> > 	Yet we would still like to clarify that our setting does not focus on the case where there are huge changes inside the pre-change LLM or post-change LLM, and we do not aim to do the root cause diagnosis, which means when temperature changes so hugely that it severely modifies the behavior of the LLM, we regard such modification as a change.
> >
> > Thank you once again for your invaluable feedback on our paper, which has motivated us to conduct a more extensive evaluation of our algorithm. We sincerely hope that we have adequately addressed your concerns and that you might consider revising your evaluation of our work.

---

> > > ### Comment · Reviewer_TL1q · 2024-11-23
> > >
> > > Thank you for your detailed response. While I appreciate the additional experiments and clarifications provided, I think you didn't answer my question about the malicious intent of watermarking and my primary concerns remain unresolved. Below, I elaborate on the key issues and provide further context for my stance:
> > >
> > > ### 1. **Motivation and Threat Model**
> > >
> > > In your response, you clarify that your focus is on developing an effective and computationally efficient online detection framework rather than diagnosing the root causes of API or model changes. First of all, I did not ask for the root cause detection or diagnosis. My concern lies in the **lack of a well-defined threat model**, which affects the motivation for the proposed detection framework.
> > >
> > > _it is complex even in white-box or traditional scenarios, and especially so in black-box LLMs with countless potential causes._ I don't think the response change is so complex that you cannot even list. For example, now I can name the following main possible reasons that lead to the response change if the user prompt is the same: 1) model weight update 2) a positive temperature 3) change of system prompt 4) decoding strategy change 5) the traffic is hijacked(there're some works like [1] based on this assumption but it's that real-world practical). However, they are either benign or unrelated to user harm.
> > > The only scenario with potential malicious intent is **system prompt changes** caused by **prompt injection attacks**, which are already well-documented (e.g., by OWASP). Even then, this represents a narrow scope of application.
> > >
> > > Regarding your focus on watermarking, I maintain my earlier point: watermarking is a well-studied method used by LLM developers to protect intellectual property. There is no evidence, to my knowledge, that applying watermarking has resulted in misleading, harmful content or sensitive data leakage, as claimed in your introduction. Without demonstrating a clear harm scenario for watermarking, the motivation for your proposed detector remains unconvincing. **Your experiments on watermarking remain heavily emphasized, yet the real-world risks posed by watermarking are not substantiated**. Without concrete examples or case studies demonstrating how watermarking leads to harm, it is unclear why users would require a detector specifically for watermark-induced changes.
> > >
> > > If the focus of your work is to address prompt injection detection (or other real-world harm you can think of raised by response change), I believe the scope and experiments require significant revision. Current evidence in Appendix E.4 is insufficient, as many advanced prompt injection scenarios (e.g., [2]) are unaddressed. Without such thorough evaluation, it is difficult to conclude that the framework is effective against prompt injection.
> > >
> > > Based on these points, I will retain my score.
> > >
> > > [1] Identifying and Mitigating Vulnerabilities in LLM-Integrated Applications
> > >
> > > [2] Formalizing and Benchmarking Prompt Injection Attacks and Defense

---

> > > > ### Author Response · Authors · 2024-11-25
> > > >
> > > > We thank the reviewer for the interaction and providing further comments and questions. We present an item-by-item response to your concerns.
> > > >
> > > > > **Q1**: Regarding the motivation: _My concern lies in the lack of a well-defined threat model, which affects the motivation for the proposed detection framework_.
> > > >
> > > > **A1**: Our detection framework is motivated by changes in language models that lead to output distribution shifts. Some changes are harmful and can cause safety concerns, which naturally necessitate a timely detection. Other changes, although not harmful, also require close monitoring and detection. Therefore, our proposed detection framework does not rely on a well-defined threat model, nor solely targets at enhancing the safety of language models.
> > > >
> > > > To better demonstrate the use cases of the detection framework, we discuss the following examples beyond safety issues. The change of output distributions is responsible for inconsistent behaviors of LLMs before and after the change. Ensuring the consistency of LLMs is significantly important in many real-world LLM-based applications such as LLM agents [1]. Meanwhile, **maintaining consistency** is essential for fostering user confidence and ensuring the reliability of the system, and **this does not necessarily involve safety**. For example, a financial assistant [2] built on a previous LLM version might fail to provide the same level of actionable advice post-update, as the updated model might produce more cautious or generic responses. Our detection method offers the user the right, or at least a means of monitoring such change and taking necessary countermeasures accordingly.
> > > >
> > > > [1] Gao, Chen, Xiaochong Lan, Nian Li, Yuan Yuan, Jingtao Ding, Zhilun Zhou, Fengli Xu, and Yong Li. "Large language models empowered agent-based modeling and simulation: A survey and perspectives." Humanities and Social Sciences Communications 11, no. 1 (2024): 1-24.
> > > > [2] Yu, Yangyang, Haohang Li, Zhi Chen, Yuechen Jiang, Yang Li, Denghui Zhang, Rong Liu, Jordan W. Suchow, and Khaldoun Khashanah. "FinMe: A Performance-Enhanced Large Language Model Trading Agent with Layered Memory and Character Design." arXiv preprint arXiv:2311.13743 (2023).
> > > >
> > > > We have also revised our introduction to clarify the scope of our study:
> > > >
> > > > “Despite their undeniable potential, the widespread adoption of LLMs has given rise to various safety, reliability, and consistency concerns (Bommasani et al., 2021; Biswas & Talukdar,
> > > > 2023). …. Throughout the paper, we term shifts of LLMs’ output distributions as changes. However, not all changes in LLM output distributions are necessarily harmful. Even benign changes, such as those introduced by LLM version updates and patches, can influence their output distributions, potentially rendering inconsistent behaviors before and after the change (Echterhoff et al., 2024). …”
> > > >
> > > > **To be continued...**

---

> > > > > ### Author Response · Authors · 2024-11-25
> > > > >
> > > > > **Cont.**
> > > > > >**Q2**: Regarding whether a benign change needs monitoring: _I don't think the response change is so complex that you cannot even list. For example, now I can name the following main possible reasons that lead to the response change if the user prompt is the same: 1) model weight update 2) a positive temperature 3) change of system prompt 4) decoding strategy change 5) the traffic is hijacked. However, they are either benign or unrelated to user harm._
> > > > >
> > > > > **A2**: Some changes, as you listed, deserve detecting and monitoring even if they do not harm users directly. These changes all lead to the potential inconsistency in responses, whose implication has been analyzed in **A1**. Such “benign” changes could happen without users' notice. For example, [Chen et al., 2023](https://arxiv.org/pdf/2307.09009) demonstrates that the performance of an LLM API can vary over time, even without official notifications of version updates. In such scenarios, an automated tool for detecting shifts in LLM behavior can be valuable, offering users additional insights and keeping them informed about potential changes. This kind of online detection tool helps users better understand and adapt to shifts in LLM capabilities, especially when they rely on the model for specific tasks that demand the consistency of the responses.
> > > > >
> > > > > >**Q3**: Regarding watermarking: _Your experiments on watermarking remain heavily emphasized, yet the real-world risks posed by watermarking are not substantiated._
> > > > >
> > > > > **A3**: In our experiments, we use watermarking not for highlighting how our algorithm can address the safety issue related to it, but for setting a change instance for evaluation. Applying some type of watermark can break such consistency and affect the performance of the system overall.
> > > > >
> > > > > >**Q4**: Regarding whether we should focus on prompt injection: _If the focus of your work is to address prompt injection detection (or other real-world harm you can think of raised by response change), I believe the scope and experiments require significant revision._
> > > > >
> > > > > **A4**: Prompt injection is an instance of change that most probably leads to safety concerns. However, we clarify that prompt injection and other real-world harm to users are **not the only focus** of this work that aims to foster better reliability of LLM. The change itself, no matter harmful or harmless, can pose performance inconsistency to a LLM. **A reliable system** should be designed to adapt to, or at least recognize changes while maintaining its intended functionality and consistency. This requires detecting and mitigating the impact of changes, in particular, including monitoring and managing benign ones that may disrupt downstream tasks or user expectations. By actively identifying such changes, we can ensure that LLM-powered applications remain robust and performant, even in dynamic or unpredictable environments.
> > > > >
> > > > > Overall, we believe our work addresses an important and timely challenge in monitoring and understanding LLM behavior. The proposed online detection framework offers a practical tool to the broader field of AI reliability and user awareness. We sincerely appreciate the reviewer’s constructive feedback, which has helped us refine and more clearly articulate the motivation and contributions of our work.

---

> > > > > > ### Comment · Reviewer_TL1q · 2024-11-25
> > > > > >
> > > > > > Thank you for your detailed response and efforts to address my concerns. While I appreciate the additional clarifications, I must reiterate that the focus and motivation of your work remain unclear and unconvincing in the context of real-world applications. Below, I further explain my concerns and suggestions for improvement.
> > > > > >
> > > > > > ### 1. Lack of Focus on Malicious Scenarios
> > > > > >
> > > > > > If your detection method is designed to address changes with malicious intent, such as system prompt tampering or prompt injection, then it could provide a meaningful contribution. However, your current focus appears to be detecting any change, regardless of its nature or impact. This broad scope dilutes the practical relevance of your work. Most potential changes (e.g., decoding updates, temperature variations, or benign model updates) are harmless and well-documented by LLM providers like OpenAI. For example, as recorded by OWASP, malicious changes like prompt injection are a real concern in LLM security, and focusing on such cases would align your work with meaningful and actionable real-world scenarios.
> > > > > >
> > > > > > On the other hand, watermarking and fingerprinting are well-studied methods in cybersecurity and ML security[1] designed to protect intellectual property. The emphasis on detecting changes caused by watermarking introduces confusion rather than clarity, as it does not align with real-world risks. Without evidence of harmful scenarios related to watermarking, this focus seems misplaced and unnecessary.
> > > > > >
> > > > > > ### 2. Misrepresentation of Notifications by LLM Providers
> > > > > >
> > > > > > You cite Chen et al., 2023 to support the claim that LLM performance can vary without official notifications of updates. However, this statement is factually incorrect. I don't know where you get this conclusion, at least based on my knowledge, OpenAI notified them. Take Chen et al., 2023 as an example, the update between March version and June version is **actually notified** by both the official news page [2] and email. This is the head of the email I received in June 13, "We're excited to announce a few updates to the OpenAI developer platform GPT-3.5 Turbo This model has been updated with a new version: gpt-3.5-turbo-0613 which is more steerable ......"This contradicts your claim that updates occur without notification. Furthermore, Chen et al., 2023 primarily measures differences in performance and alignment, which your detector is not designed to address. These inaccuracies weaken the justification for your work and should be rectified.
> > > > > >
> > > > > > ### 3. Practical Necessity of General Change Detection
> > > > > >
> > > > > > Even if benign changes cause response inconsistencies, it remains unclear why a general-purpose change detector is necessary. Many benign changes, such as decoding updates or model fine-tuning, do not harm users and are often communicated transparently by providers. While consistency is valuable, it is unclear why a detection tool is needed to monitor such changes when they do not impact user safety or performance in critical tasks.
> > > > > >
> > > > > > If your aim is to address benign changes, I encourage you to provide compelling examples where such changes critically impact downstream applications. For example, financial applications requiring strict consistency might benefit from detecting certain changes, but this would require specific and rigorous experiments to justify your approach.
> > > > > >
> > > > > > Until these issues are addressed, the necessity and relevance of your proposed detection framework remain unconvincing. I hope these points help refine your work and improve its alignment with practical and impactful scenarios.
> > > > > >
> > > > > > [1] Protecting Intellectual Property of Machine Learning Models via Fingerprinting the Classification Boundary
> > > > > >
> > > > > > [2] https://openai.com/index/function-calling-and-other-api-updates/

---

> > > > > > > ### Author Response · Authors · 2024-11-30
> > > > > > >
> > > > > > > We thank the reviewer for further clarification and comments. We try to address your concerns in two parts.
> > > > > > >
> > > > > > > > **Q1&Q3**: Regarding our focus on general changes.
> > > > > > >
> > > > > > > **A1**:  The proposed method is designed as a general detection framework rather than a specific method targeting particular types of change. While we agree that changes of benign intention and malicious intention may have different levels of safety concerns, we maintain that all changes, regardless of their intent, should be detected. On the one hand, LLM reliability is not a concept that only focuses on potential harmful outputs of a LLM. It also encompasses the model's consistency and predictability in delivering expected performance across different contexts and updates, which can be disrupted by even benign changes. On the other hand, even if in certain cases benign changes do not pose a direct threat to most users, we still assert that in a context where human rights are highly valued, every user should at least have the right to be informed, since for some users with specific needs, such changes may impact the reliability of outputs, directly affecting their applications. Moreover, the importance of watermarking detection is evident in the extensive literature, we give a few examples below:
> > > > > > >
> > > > > > > [1] Li, Xiang, Feng Ruan, Huiyuan Wang, Qi Long, and Weijie J. Su. "A Statistical Framework of Watermarks for Large Language Models: Pivot, Detection Efficiency and Optimal Rules." Annals of Statistics, to appear.
> > > > > > >
> > > > > > > [2] Yang, Xianjun, Liangming Pan, Xuandong Zhao, Haifeng Chen, Linda Petzold, William Yang Wang, and Wei Cheng. "A survey on detection of LLMs-generated content." arXiv preprint arXiv:2310.15654 (2023).
> > > > > > >
> > > > > > > [3] Chakraborty, Souradip, Amrit Singh Bedi, Sicheng Zhu, Bang An, Dinesh Manocha, and Furong Huang. "On the possibilities of ai-generated text detection." arXiv preprint arXiv:2304.04736 (2023).
> > > > > > >
> > > > > > > [4] Li, Xingchi, Guanxun Li, and Xianyang Zhang. "Segmenting Watermarked Texts From Language Models." In The Thirty-eighth Annual Conference on Neural Information Processing Systems, 2024.
> > > > > > >
> > > > > > >  A use case of watermark detection in the online setting is to distinguish the AI-generated content as soon as possible. Quick and accurate watermark detection helps preserve the credibility of digital media, and helps keep people informed about AI-generated texts. While various watermarking schemes exist, a fundamental requirement is that the watermarks must be detectable. Therefore, our proposed general detection algorithm was validated by detecting watermarks in such an online manner and was able to identify watermarked text quickly and efficiently.
> > > > > > >
> > > > > > > Compared to other methods specifically designed to detect malicious usage or attacks, which often come with higher costs and efforts, our approach provides a cost-effective means of identifying a "necessary condition" for the occurrence of changes. It can also serve as a preliminary screening tool for detecting anomalies; if suspicious signals are flagged, users can then opt for more targeted and expensive anti-attack methods for further investigation.
> > > > > > >
> > > > > > >
> > > > > > > > **Q2**: Regarding LLM API updates.
> > > > > > >
> > > > > > > **A2**: We are aware that updates in LLM API are usually announced by its providers, and our experiments in real-world LLM APIs clearly detected such updates. However, we also believe that not all changes are announced. In our experiments in LLM APIs, from June 01 to August 31, we observed changes in GPT-4-turbo and Jamba. The change in GPT-4-turbo is located in around July 23 to July 29, but is not presented in [OpenAI changelog website](https://platform.openai.com/docs/changelog). Also, the change in Jamba, detected at around June 27, is not documented in [AI21 changelog](https://docs.ai21.com/changelog).
> > > > > > >
> > > > > > > In [this paper: How Is ChatGPT’s Behavior Changing over Time?](https://arxiv.org/pdf/2307.09009), the authors suggested that _“...These unknowns make it challenging to stably integrate LLMs into larger workflows: if LLM’s response to a prompt (e.g. its accuracy or formatting) suddenly changes, this might break the downstream pipeline. It also makes it challenging, if not impossible, to reproduce results from the “same” LLM.”_ Though the authors acknowledged that their evaluations were taken on March and June version of GPT, they highlighted “the need to continuously evaluate and assess the behavior of LLM drifts in applications”, because nobody can guarantee what a change will happen in the LLM.
> > > > > > >
> > > > > > > We sincerely appreciate your thoughtful review and the opportunity to address your concerns. We hope that we have conveyed the rationale behind our approach – ensuring the reliability and transparency of these systems for all users. We believe our work contributes meaningfully to this objective. We look forward to further discussion with you on this perspective. Thank you again for your time and consideration.

---

> > > > > > > > ### Comment · Reviewer_TL1q · 2024-11-30
> > > > > > > >
> > > > > > > > Dear Authors,
> > > > > > > >
> > > > > > > > Thank you for your continued engagement and detailed responses to my concerns. I appreciate your efforts to clarify your motivations and provide additional context. So, we are at least now on the same page that OpenAI GPT for March and June in 2023 is notified, right? Actually most of recorded major changes is by OpenAI Dev day and informed by email. Now let's keep the discussion going.
> > > > > > > >
> > > > > > > > ### **1. Observed Unconfirmed Changes in GPT-4-turbo**
> > > > > > > >
> > > > > > > > You mentioned detecting changes in GPT-4-turbo between July 23 and July 29, 2024 (I don't use Jamba at all, so let's focus on OpenAI). As per my knowledge, the latest version available during that period was `gpt-4-turbo-2024-04-09`. Although you don't care about the reason of change, but I care, so let me analyze for you. Let's analyze the possible reasons: 1) OpenAI ninja edited the model without notifying users 2)Elevated error 3)false positives by your detetcor.
> > > > > > > >
> > > > > > > > Let's first check reason 1), I searched the https://community.openai.com/ to find relevant reports, but found none for GPT-4-turbo. There is one thread complaining performance degrade: https://community.openai.com/t/has-regular-gpt-4-model-changed-for-the-worse-by-any-chance/804058, but it is for gpt-4-1106-preview and located in June 5. Neither model nor date matches your findings. For reason 2), I checked the https://status.openai.com/history?page=2, again, neither model name nor date mathes your results. I may miss some, so please let me know if you have extra evidences. Since there is no time machine back to that date, I can only lean towards the reason that it is a false positive due to  ``Temperature is non-zero throughout all our experiments`` as you acknowledged.
> > > > > > > >
> > > > > > > > One potential reason for these detections could be the use of a non-zero temperature in your experiments. As you acknowledged, you are using a higher temperature (1.0) which could introduce randomness into the model's outputs, which might lead to perceived changes that are artifacts of stochastic variation rather than actual model updates.
> > > > > > > >
> > > > > > > > To illustrate this point, I conducted a simple experiment using OpenAI's API with the temperature set to zero:
> > > > > > > >
> > > > > > > > ```python
> > > > > > > > import openai
> > > > > > > >
> > > > > > > > openai.api_key = "YOUR_API_KEY"
> > > > > > > >
> > > > > > > > completion = openai.ChatCompletion.create(
> > > > > > > >     model="gpt-4-turbo-2024-04-09",
> > > > > > > >     messages=[
> > > > > > > >         {"role": "user", "content": "Write the 8 queens problem in Python."}
> > > > > > > >     ],
> > > > > > > >     temperature=0
> > > > > > > > )
> > > > > > > >
> > > > > > > > print(completion.choices[0].message.content)
> > > > > > > >
> > > > > > > > ```
> > > > > > > >
> > > > > > > > With `temperature=0`, the outputs are deterministic, making it straightforward to detect any changes:
> > > > > > > >
> > > > > > > > -   **`gpt-4-turbo-2024-04-09` Output:**
> > > > > > > >
> > > > > > > >     ```
> > > > > > > >     The 8 Queens problem is a classic example of a constraint satisfaction problem that can be solved using backtracking. The goal is to place 8 queens on an 8x8 chessboard...
> > > > > > > >
> > > > > > > >     ```
> > > > > > > >
> > > > > > > > -   **Earlier Versions Output:**
> > > > > > > >
> > > > > > > >     -   **`gpt-4-0125-preview`:**
> > > > > > > >
> > > > > > > >         ```
> > > > > > > >         The 8 Queens problem is a classic example of a constraint satisfaction problem that involves placing eight queens on an 8x8 chessboard so that no two queens threaten each other...
> > > > > > > >
> > > > > > > >         ```
> > > > > > > >
> > > > > > > >     -   **`gpt-4-1106-preview`:**
> > > > > > > >
> > > > > > > >         ```
> > > > > > > >         The 8-queens problem is a classic puzzle in which the goal is to place eight queens on an 8x8 chessboard in such a way that no two queens threaten each other. This means that no two queens can be in the same row, column, or diagonal...
> > > > > > > >
> > > > > > > >         ```
> > > > > > > >
> > > > > > > >
> > > > > > > > These differences are readily apparent and can be detected without complex methods. This raises the question of whether using high-temperature settings might artificially complicate the detection task. It seems that setting a zero temperature could simplify the problem and make detection more reliable. If I see such change during July 23 to July 29, then there must be changes to the model or default API setting, even if OpenAI does not inform that, I am sure there must be changes. However, for your results, it's not convincing for me to believe.

---

> > > > > > > > > ### Comment · Reviewer_TL1q · 2024-11-30
> > > > > > > > > **Continued**
> > > > > > > > >
> > > > > > > > > ### **2. Emphasis on Watermarking Detection**
> > > > > > > > >
> > > > > > > > > You provided several references on watermarking in LLMs to justify the importance of detecting such changes, but I wonder if you really read them before giving them to me as evidence. Unfortunately, the watermarking schemes discussed in these papers typically involve embedding identifiable patterns into generated text, often using a secret key known to the verifier. The primary goal is to attribute content to a specific model or protect intellectual property.
> > > > > > > > >
> > > > > > > > > For example:
> > > > > > > > >
> > > > > > > > > -   **Li et al., "A Statistical Framework of Watermarks for Large Language Models":** This work assumes access to a secret key for watermark verification.
> > > > > > > > > -   **Yang et al., "A Survey on Detection of LLMs-Generated Content":** The paper states that ``Watermarking is designed to determine whether the text is coming from a specific language model rather than universally detecting text generated by any potential model.``
> > > > > > > > >
> > > > > > > > > These watermarking detection methods differ significantly from your setting, where you aim to detect general changes in output distributions without access to such keys or patterns. I suggest reading this paper[1] to have a clear understanding of watermarking in LLM. Take one step back, if you really want to position yourself as watermarking detection, why don't you cite these papers in your paper and compare with them?
> > > > > > > > > [1] Instructional Fingerprinting of Large Language Models
> > > > > > > > >
> > > > > > > > > ### **3. Reference to "How Is ChatGPT’s Behavior Changing over Time?"**
> > > > > > > > >
> > > > > > > > > Regarding the paper by Chen et al., while the authors highlight the importance of assessing LLM drifts, they focus on performance metrics like accuracy and reasoning abilities, not on output variability due to stochastic processes. If you take a further look at their paper, you will find `We set the temperature to be 0.1 to reduce output randomness, as creativity was not needed in our evaluation tasks.` in their paper. So, why don't him take temperature as 0 for deterministic results? If you take a look at their paper and github repo https://github.com/lchen001/LLMDrift/blob/main/generation/LEETCODE_EASY_EVAL.csv, you will find that they are using pass@4 as accuracy. So that makes sense. However, even though, they are setting a such a low temperature
> > > > > > > > >
> > > > > > > > > In your experiments, you use a temperature of 1.0, which introduces significant randomness into the outputs. This high temperature may obscure the detection of genuine model changes versus variability due to the sampling process.
> > > > > > > > >
> > > > > > > > > ### **4. Practicality of Your Detection Method**
> > > > > > > > >
> > > > > > > > > Given that simpler methods can effectively detect changes when the temperature is set to zero or a low value, it is unclear why a more complex detection framework is necessary for practical applications. Introducing high temperatures seems to create additional challenges that may not reflect real-world usage scenarios, especially in contexts where consistency and reliability are paramount.
> > > > > > > > >
> > > > > > > > > ### **Conclusion**
> > > > > > > > >
> > > > > > > > > While I appreciate your goal of developing a general detection framework, I believe that the current approach may not adequately address the practical challenges or demonstrate clear advantages over simpler methods. By refining your focus and adjusting your experimental design, your work could make a more substantial contribution to the field.

---

> > > > > > > > > > ### Author Response · Authors · 2024-12-03
> > > > > > > > > > **Further responses to Reviewer TL1q**
> > > > > > > > > >
> > > > > > > > > > We thank the reviewer for further clarification and comments.
> > > > > > > > > >
> > > > > > > > > > > **Q1**: Regarding _Observed Unconfirmed Changes in GPT-4-turbo_
> > > > > > > > > >
> > > > > > > > > > **A1**: As you have found out, for GPT4-turbo, we observed a surge in detection statistics around July 24 (between 23 and 29 July) that has not been documented in openai’s changelog. We mentioned this potential change in Section 4.2 and provided the trajectory of detection statistics of **six prompts** to support this conjecture in Figure 17 (see Appendix). We have 14 prompts in this real world detection, and most of them witnessed the increase of detection statistics. Thus we believe it is not so likely that our prompts can raise the false alarm independently and simultaneously on such a certain date. However, since the ground truth for such surge is uncertain, we can only conjecture, as stated in the paper, that there may be changes in the model itself.
> > > > > > > > > >
> > > > > > > > > > > **Q2**: Regarding non-zero temperature
> > > > > > > > > >
> > > > > > > > > > **A2**: We agree that when the temperature is set to 0, the responses become deterministic and the detection becomes much simpler. However, we woud like to clarify our temperature choice in the work. First, according to public knowledge, the default temperature of mainstream website chatbot services is set to 0.7 or 1.0 (it’s not publicated by OpenAI, but we can found discussion like [this](https://community.openai.com/t/web-chat-default-temperature-for-gpt-3-5-and-4/167356/6)), far larger than 0.0, to instill variation of responses to even the same prompt, serving user’s need for diversity. Thus we set the temperature in our setting to 1.0 rather than 0.0 is **out of practical consideration**. The experiments show that our algorithm has a good performance when temperature is 1.0 and responses are stochastic, which is much harder compared to setting the temperature = 0.0. The proposed method will still perform well (and much better) for a zero or lower temperature. Furthermore, as you suggested in your first official comment, we also did extra experiments when temperature can vary in a certain interval (see Figure 21) to address your concern on the **robustness of the algorithm**. Based on these empirical evidences, our proposed detection method works well for a wide range of temperature values and thus remains effective and aligns better with practical scenarios.
> > > > > > > > > >
> > > > > > > > > > > **Q3**:  Regarding watermarking
> > > > > > > > > >
> > > > > > > > > > **A3**: We understand your concern and we are well aware of the concept of “watermark detection”. In our paper, we didn't name the synthetic change scenario as “watermark detection” in the classical sense (which is an offline hypothesis testing problem to distinguish watermarked versus unwatermarked data), but as “emergence of watermark” in our sequential setting. As we stated before, we aimed to use this _controllable_ case of an artificial change of response distribution to show how sensitive our algorithm is to a change. With watermark, we can easily set the strength of the distribution change by setting the watermarking parameters, and thus to further study the detection efficiency v.s. fase alarm trade-off under different change strengths. This analysis is barely possible when we only focus on the version update or prompt injection scenario.
> > > > > > > > > >
> > > > > > > > > > We would like to thank you once again for your continued engagement and the effort you have put into reviewing our work. We sincerely appreciate your thoroughness and dedication throughout this process and hope that our clarifications have provided a better understanding of our contributions. We kindly ask you to consider the broader implications and significance of our work when making your final decision.

---

> > > > > > > > > > > ### Comment · Reviewer_TL1q · 2024-12-03
> > > > > > > > > > >
> > > > > > > > > > > Dear Authors,
> > > > > > > > > > >
> > > > > > > > > > > Thank you for your reply. While I appreciate your reply, I must reiterate that my main concerns remain unaddressed.
> > > > > > > > > > >
> > > > > > > > > > > ### **1. Lack of Concrete Evidence for Detected Changes in GPT-4-turbo**
> > > > > > > > > > >
> > > > > > > > > > > You mention observing changes in GPT-4-turbo between July 23 and July 29 that are not documented by OpenAI. Without concrete and deterministic evidence, it is unconvincing to conclude that OpenAI made stealth updates to a model that is officially stated to have its last update on 2024-04-09 (`gpt-4-turbo-2024-04-09`).
> > > > > > > > > > >
> > > > > > > > > > > Moreover, using a non-zero temperature in your experiments introduces randomness into the model's outputs, which can lead to false positives in detecting changes. As I demonstrated earlier with code snippets using `temperature=0`, deterministic outputs allow for straightforward detection of actual changes. I would believe there would be some model/API change if you are using zero temperature, but unfortunatly there's not. This suggests that the changes your method detected may be due to stochastic variations rather than genuine model updates.
> > > > > > > > > > >
> > > > > > > > > > > ### **2. Justification for Using Non-Zero Temperature**
> > > > > > > > > > >
> > > > > > > > > > > You argue that setting the temperature to zero is impractical because mainstream chatbot services use higher default temperatures to introduce variation in responses. However, all popular closed-source LLM APIs, including OpenAI, Claude, Gemini, Qwen,Mistral, allow users to set the temperature parameter. Users who require consistency in detection can and do set the temperature to zero to obtain deterministic detection outputs. Therefore, it is both practical and feasible to use a zero temperature setting for detecting changes.
> > > > > > > > > > >
> > > > > > > > > > > Even in the paper "How Is ChatGPT’s Behavior Changing over Time?" that you referenced, the authors set the temperature to 0.1 to reduce output randomness for their evaluation tasks. It is not 0 because the author needs the pass@4 accuracy. This choice highlights the importance of minimizing randomness when assessing model behavior over time.
> > > > > > > > > > >
> > > > > > > > > > > By choosing a high temperature (e.g., 1.0), you introduce significant variability into the outputs, which complicates the detection task unnecessarily. This approach may create an artificially complex problem that your method aims to solve, whereas simpler methods suffice under more controlled settings.
> > > > > > > > > > >
> > > > > > > > > > > ### **3. Focus on Watermarking Contradicts Claim of General Change Detection**
> > > > > > > > > > >
> > > > > > > > > > > You state that your method is designed for general change detection, yet much of your work focuses on watermarking as a means to introduce controllable changes. This focus seems contradictory. If your goal is to detect general changes, it would be more appropriate to consider a variety of controllable response changes that are more directly relevant, such as modifying the system prompt, adjusting decoding strategies, or altering other parameters.
> > > > > > > > > > >
> > > > > > > > > > > Moreover, the emphasis on watermarking—especially when watermarking is not a standard practice in LLM APIs and is primarily used to protect intellectual property—does not convincingly demonstrate the practical necessity of your method. There are more direct methods to create controllable changes for evaluation purposes.
> > > > > > > > > > >
> > > > > > > > > > > Overall, my primary concerns about the practical necessity and real-world applicability of your detection framework remain unaddressed.

---

> > > > > > > > > > > > ### Author Response · Authors · 2024-12-04
> > > > > > > > > > > >
> > > > > > > > > > > > About temperature:
> > > > > > > > > > > > We would like to further clarify that our proposed method is effective across a wide range of temperature values. In particular, we demonstrated a favorable tradeoff between detection efficiency and false alarm rates, even with temperature settings of 0.7 or 1.0. While the reviewer mentioned the possibility of setting the temperature to 0, we note that: (1) reducing the temperature to 0 significantly limits the diversity of generated responses, which we believe is essential and meaningful for practical applications; and (2) the temperature parameter is not always user-configurable, as is the case with platforms like Microsoft Copilot. Therefore, we believe a detector that can handle non-zero temperatures (also trivially applicable to zero temperatures) offers more practical usage.
> > > > > > > > > > > >
> > > > > > > > > > > > We would also like to reiterate that we did not assert whether the detected “change” in GPT-4-turbo between July 23 and July 29 corresponds to a real change or a false alarm. Our statement was simply an observation of a surge in the detection statistics. This aligns with a well-known challenge in real-data online detection: the ground-truth change time and scenario may be unknown in most real-world cases. As a result, evidence from detection statistics typically does not lead to a definitive conclusion about the presence of an actual change. Instead, such evidence is very useful since it is indicative, providing users with a signal so that they may choose to take preventive actions.

---

### Official Review · Reviewer_niaS · 2024-11-04

**Soundness:** 3
**Presentation:** 2
**Contribution:** 3
**Rating:** 6
**Confidence:** 3

**Summary:**

The paper presents a novel online change detection method for black-box Large Language Models (LLMs), combining a CUSUM-type detection statistic based on entropy and Gini coefficient with a UCB-based adaptive prompt selection strategy to effectively detect changes in LLMs. The experimental results demonstrate the method's effectiveness on both synthetic and real-world APIs, covering various change scenarios.

**Strengths:**

Innovation: The paper proposes an online change detection approach tailored for black-box LLMs, which is a relatively new research direction and shows high innovation.
Methodology: The combination of CUSUM-type detection statistics and UCB strategy enhances the accuracy and efficiency of change detection.
Experimental Design: The validation of the method on both synthetic and real-world data, including different change scenarios, strengthens the paper's credibility.
Practical Application: The method is significant for ensuring the stability and reliability of LLMs in commercial and safety-critical applications.

**Weaknesses:**

Generalizability: The paper could benefit from additional experiments to demonstrate the generalizability of the method across more datasets and application scenarios.
Computational Resources: While the paper mentions considerations for computational resources, a deeper discussion on optimizing the algorithm for resource-constrained environments would be beneficial.
Adversarial Attacks: The paper does not address the robustness of the detection system against adversarial attacks designed to evade detection.

**Questions:**

Generalizability: Could the authors provide more experimental results on the generalizability of the algorithm across different types and scales of LLMs?
Adversarial Attacks: Have the authors considered the impact of adversarial attacks on the detection algorithm, and how could the robustness against such attacks be improved?
Real-time Performance: Could the authors provide data on the performance of the algorithm in real-time or near-real-time environments?
Computational Efficiency: Could the authors further discuss possible methods for optimizing the computational efficiency of the algorithm in resource-constrained settings?

---

> ### Author Response · Authors · 2024-11-20
>
> Dear reviewer niaS,
>
> We thank the reviewer for the comments, and we appreciate the time you spent on the paper. Below we address the concerns and comments that you have provided.
>
> > **Q1**: Regarding generalizability
>
> **A1**: We have added two additional experiments in the revised version:
> (1) **Detecting prompt injection**: Based on the suggestion from other reviewers, we have tested our algorithm's performance in detecting prompt injection scenarios, and it also performs effectively in this context (see Appendix E.4 in the revised draft).
> (2) **Detecting Model Changes with Variable Temperatures**: We have also included experiments to detect model changes when the temperatures of the pre- and post-change models both vary within a specified range (in contrast to our previous results, where the temperature was held constant). Our detection method continues to perform well in this scenario, illustrating its robustness, especially when responses may come from distributions that differ due to varying temperatures within the same model (see Appendix E.3 in the revised draft).
>
> We hope these additional experiments provide further evidence of our method’s robustness and performance, alongside our previous results demonstrating detection across diverse scenarios, including watermark emergence, synthetic version updates, and real-world version updates.
>
>
> > **Q2**: Regarding adversarial attack study
>
> **A2**: To demonstrate the robustness of our algorithm (which may partially address your concern), we conducted additional experiments in which the temperature of both the pre-change and post-change LLMs varied, uniformly sampled from the range [0.9, 1.0]. And in this case, our proposed detection algorithm continued to perform effectively without significant degradation, as shown in the **newly added Figure 21**.
>
> Furthermore, we would like to emphasize that, to the best of our knowledge, this work is the first to formulate the online detection problem for black-box LLMs and establish a solution that demonstrates strong performance and high efficiency. While we acknowledge the validity of your concern about adversarial attacks, this falls outside the current scope of our study, though it presents an interesting direction for future research. The current detection framework could be extended to robust counterparts by incorporating distributional uncertainties introduced by adversarial attacks. For instance, with prior knowledge of the type of adversarial attack, we could frame the online detection problem as a distributionally robust change detection task, aiming to detect changes from a pre-change ambiguity set of response distributions to a non-overlapping post-change set. The sizes of these sets would reflect the strength of the adversarial attack. Nonetheless, this extension is beyond the current study’s focus.
>
>
> > **Q3**: Regarding real-time performance
>
> **A3**:  In our study, we highlight the real-time setting of online detection throughout the paper, and all results are obtained under the real-time detection environment. Specifically, in our real-world dataset, we collect data on a daily basis, so that we can make daily decision on whether to report a change based on the responses collected so far. In our synthetic experiments, even though we collect data before we conduct the detection algorithm, we always run the detection procedure in an **online** manner, which means that each decision about whether to raise alarm is completely based on the information collected by that time step.
>
>
> **To be continued...**

---

> > ### Author Response · Authors · 2024-11-20
> >
> > **Cont.**
> >
> > > **Q4**: Regarding computational efficiency
> >
> > **A4**:  We fully agree that computational efficiency is a crucial factor, especially in the online setting where real-time decision-making is desired. Addressing this concern has been a key focus in designing our detection method. In our proposed detection procedure, there are several steps to improve computational efficiency:
> > 1. **Recursive update in detection statistics**: We use cumulative statistics that can be computed in a recursive way to conduct the detection procedure, which allows us to update the detection statistics based on newly collected data merely with the detection statistics at the previous time step. We do not suffer from the huge cost of computing statistics using all the data collected before current time. And this update method is already the cost-efficient one in detection context.
> > 2. **Calculation of N-token metrics**: To calculate N-token metrics, we effectively addressed the issue of exponential growth of different combinations of N-tokens by an approximation method (see Section 3.2 in our paper). This method improved the computational efficiency and sample efficiency by merging N tokens into one set to avoid the need of sufficient sampling on the combination space of N tokens.
> > 3. **Dynamic prompt selection**: We are fully aware of the different sensitivity of prompts to a change, and thus we adapted UCB algorithm into our detection framework, which requires less queries (eg. 5 out of 14) each time and maintains strong performance as the best prompt.
> > 4. **Memory-efficient framework**: Due to the recursive nature of our detection statistics, the information from previous time steps is condensed into four detection statistics, while the new information at each time step is similarly converted into four metrics. As a result, the detection process is memory-efficient, requiring storage of only a limited set of statistics.
> >
> > Further improvement: We need to emphasize that the computational cost of the current algorithm already satisfies the need for real-time detection in LLM. Further improvements could involve designing prompts with higher sensitivity based on the current well-performing prompts. We will explore this in future studies.
> >
> > Thank you again for your comments on our paper, which have guided us toward a more refined revision. We sincerely hope that we have addressed your concerns, and that you may consider improving your evaluation of our work.

---

> > > ### Author Response · Authors · 2024-11-25
> > > **Follow-Up on Reviewer Feedback**
> > >
> > > Dear Reviewer niaS,
> > >
> > > We hope this message finds you well. We are following up regarding our paper during the rebuttal phase. We have carefully addressed your concerns and made improvements to the paper accordingly.
> > >
> > > Since we are approaching the end of the discussion period, we kindly ask if your concerns have been adequately addressed by our rebuttal and revision. If so, we would greatly appreciate it if you could consider updating your evaluation. If there are any remaining concerns, we are happy to follow-up.
> > >
> > > Thank you for your time and consideration.
> > >
> > > Best regards,
> > > Authors 13317

---

> ### Author Response · Authors · 2024-12-01
> **Follow-Up on Paper 13317 - Revision**
>
> Dear Reviewer niaS,
>
> We hope this message finds you well. We are writing to follow up on our paper during the rebuttal phase. In response to your feedback, we have conducted additional experiments and made significant revisions to the paper, which we believe address your concerns.
>
> As we near the conclusion of the discussion period, we kindly ask if you feel that our rebuttal and the updated paper adequately resolve your points. If so, we would greatly appreciate it if you could update your evaluation. Of course, if there are any remaining issues or areas needing clarification, please let us know—we would be happy to provide further details.
>
> Thank you very much for your time and valuable input.
>
> Best regards,
> Authors 13317

---

> > ### Author Response · Authors · 2024-12-03
> > **Follow-Up on Paper Rebuttal**
> >
> > Dear Reviewer niaS,
> >
> > We hope this message finds you well. As the discussion period is quickly coming to a close, we wanted to follow up regarding our paper. In response to your feedback, we have conducted additional experiments and made significant revisions, which we believe address your concerns.
> >
> > Could you kindly let us know if our rebuttal and the updated paper sufficiently resolve your points? If so, we would greatly appreciate it if you could update your evaluation at your earliest convenience.
> >
> > Thank you again for your time and invaluable feedback!
> >
> > Best regards,
> > Authors 13317

---

### Author Response · Authors · 2024-11-20
**Responses to all the reviewers**

For all reviewers:

We thank the reviewers for their valuable feedback and constructive suggestions.

**Experiments added**:
- Prompt Injection Detection: We have added an experiment to test our algorithm’s effectiveness in detecting prompt injection scenarios. See new details in **Appendix E.4 in the revised draft**.
- Varying Model Temperature: We conducted additional tests with varying model temperatures, rather than a fixed constant temperature, to demonstrate robustness under fluctuating conditions. See **Appendix E.3**.
- Comparison with a Simple Baseline: We compared our approach with a straightforward baseline as suggested by the reviewer based on similarity measures. Details of this comparison can be found in **Appendix E.1**.

**Clarifications added**:
- Experimental Details: We clarified our use of a fixed, non-zero temperature in all previous experiments. Added specifics can be found in Appendix C paragraph 1 in the revised draft.
- Model Access Assumptions: We clarified that our approach assumes **no** access to top-K probability distributions, which is consistent with limitations in most available models. This assumption is intrinsic in our black-box LLM setting.
- Real-Time Detection Setup: We have specified that our method is designed for online detection, i.e. make decisions in real time, as noted in our title, and line 83 in introduction, and paragraph starting from line 165 in problem setup.
- Evaluation on Larger LLMs: We evaluated our algorithm on vicuna-7B, as well as real-world APIs (whose parameter size is in trillions).

---

### Meta-Review · Area_Chair_xASL · 2024-12-22

**Metareview:**

The reviewers were split about this paper and did not come to a consensus: on one hand they appreciated the paper clarity, on the other they had concerns with (a) the overall motivation of the work (b) the use of watermarking in experiments. Three reviewers responded to the author feedback (wJQu, with a short response, EvAs with two sentences indicating that they would maintain their score, and TL1q, with extremely detailed feedback). No reviewers engaged in further discussion of the paper. After going through the paper and the discussion I have decided to vote to reject based on the above issues. Specifically, for (a) a reviewer argued that the authors should have motivated their paper by a malicious setting where the model is being attacked, instead of arbitrary change detection. The authors responded that any change should be detected in order to guarantee LLM reliability. Both author and reviewer continued to argue these points and there was no resolution. The crux of the authors’ argument is that detecting any change is important as “maintaining consistency is essential for fostering user confidence and ensuring the reliability of the system”. They give an example of a financial assistant. I agree with the authors that there may be similar high-risk scenarios where any change to the model should be detected. However, the attackability and instability of current, and likely, future LLMs means that they will not be used in these settings for some time, even with an any-change detector. For scenarios outside of this, this is overkill. Models are changed frequently and often in very small ways. This does not mean that an important change will not occur, but a method that detects any change is not informative of this, but requires additional tools to assess the importance of the change, e.g., reliability evaluations. In this case, if external evaluations are being used any time the any-change detector triggers, the benefit of an any-change detector is the computation saved by running the external evaluation. This is what should have been evaluated by the authors, but this is not included. For (b) a reviewer took issue with the use of watermarking as an example use-case for any-change detection. The authors argued that this was used to be controllable as the strength of the distribution shift can be changed by setting watermark parameters. They also argued that the reviewer was convinced that “watermarking should only be applied for its traditional purpose has led to an unwarranted critique”. This is an unfair description of the reviewer’s position. Their argument was that your experiments relied heavily on this setup, that this setup is artificial, and that there are other, more realistic ways to go about controllable LLM changes. The reviewer is correct. You could have fine-tuned a model towards a target, controlling the number of update steps, in order to produce controllable LLM changes. You could have altered decoding parameters. Both of these are much closer to your motivation than altering watermarking parameters. Given all of the above, I believe this work should be rejected at this time. Once these things and other issues mentioned in the reviews are addressed in an updated version, the work will be much improved.

**Additional Comments On Reviewer Discussion:**

Please see the metareview for these details.

---

### Decision · Program_Chairs · 2025-01-22

Reject